# The cholesterol transport protein GRAMD1C regulates autophagy initiation and mitochondrial bioenergetics

Matthew Yoke Wui Ng [1,2], Chara Charsou[1,2], Ana Lapao [1,2], Sakshi Singh [1,2], Laura Trachsel-Moncho [1,2], Sebastian W. Schultz [2,3], Sigve Nakken [2,4], Michael J. Munson [1,2,5] & Anne Simonsen [1,2,3] ✉

During autophagy, cytosolic cargo is sequestered into double-membrane vesicles called autophagosomes. The contributions of specific lipids, such as cholesterol, to the membranes that form the autophagosome, remain to be fully characterized. Here, we demonstrate that short term cholesterol depletion leads to a rapid induction of autophagy and a corresponding increase in autophagy initiation events. We further show that the ER-localized cholesterol transport protein GRAMD1C functions as a negative regulator of starvation-induced autophagy and that both its cholesterol transport VASt domain and membrane binding GRAM domain are required for GRAMD1C-mediated suppression of autophagy initiation. Similar to its yeast orthologue, GRAMD1C associates with mitochondria through its GRAM domain. Cells lacking GRAMD1C or its VASt domain show increased mitochondrial cholesterol levels and mitochondrial oxidative phosphorylation, suggesting that GRAMD1C may facilitate cholesterol transfer at ER-mitochondria contact sites. Finally, we demonstrate that expression of GRAMD family proteins is linked to clear cell renal carcinoma survival, highlighting the pathophysiological relevance of cholesterol transport proteins.

Macroautophagy (referred henceforth as autophagy) involves the de novo formation of membranes that sequester cytoplasmic cargo into double-membrane autophagosomes, which subsequently fuse with lysosomes, leading to cargo degradation and recycling of the resulting macromolecules to obtain homeostasis during periods of starvation and cellular stress. Autophagosome biogenesis is initiated through the recruitment of the ULK1 kinase complex and the class III phosphatidylinositol 3-kinase complex 1 (PIK3C3-C1) to the endoplasmic reticulum (ER)-associated sites, from where newly formed autophagosomes emanate. The autophagosome membrane is largely devoid of transmembrane proteins[1] and its formation is therefore thought to be regulated by membrane-associated proteins, as well as its lipid composition and lipid distribution[2]. Previous studies have shown that autophagosomes are enriched in unsaturated fatty acids[3,4] and that these are necessary for autophagosome formation[5,6]. These observations suggest that decreased membrane order is favorable towards autophagy initiation, possibly by generating flexible, highly curved membranes known to be required for autophagosome formation[7]. In addition to phospholipids, cholesterol is a crucial component of mammalian membranes, and its abundance is also a determinant of membrane order and fluidity[8,9]. It is noteworthy that freeze-fracture electron microscopy analysis revealed early autophagosomal

[1]Department of Molecular Medicine, Institute of Basic Medical Sciences, University of Oslo, 0372 Oslo, Norway. [2]Centre for Cancer Cell Reprogramming, Institute of Clinical Medicine, University of Oslo, 0450 Oslo, Norway. [3]Department of Molecular Cell Biology, Institute for Cancer Research, Oslo University Hospital Montebello, 0379 Oslo, Norway. [4]Department of Tumor Biology, Institute for Cancer Research, Oslo University Hospital Montebello, 0379 Oslo, Norway. [5]Present address: Advanced Drug Delivery, Pharmaceutical Sciences, BioPharmaceuticals R&D, AstraZeneca, Gothenburg, Sweden. ✉e-mail: anne.simonsen@medisin.uio.no

structures to be cholesterol poor[10], suggesting that cholesterol-poor membranes are the principal source of membranes during autophagosome biogenesis. In agreement with this, cholesterol depletion with methyl-β cyclodextrin (MBCD) and statins have been reported to promote LC3 lipidation and its turnover[11–16]. A few studies have however found that high cholesterol levels promote autophagy[17,18]. The majority of these studies involved long-term cholesterol manipulation that can lead to metabolic rewiring, and changes in transcriptional and signaling pathways[19] and might therefore not reflect the direct influence of the cholesterol in autophagy.

Intracellular cholesterol levels are maintained through a combination of biogenesis and extracellular cholesterol uptake. In addition to the low-density lipoprotein receptor (LDLR) pathway, several cholesterol transport proteins have been shown to mediate import and intracellular cholesterol transport directly from the plasma membrane (PM)[20,21]. An example of such cholesterol transport proteins belongs to the GRAMD family proteins (consisting of GRAMD1A, GRAMD1B, GRAMD1C (also known as Aster Proteins), GRAMD2, and GRAMD3), named after the lipid-binding PH-like GRAM domain in their N-terminal region. Among the five members, only GRAMD1A, GRAMD1B and GRAMD1C (herein collectively referred to as GRAMD1s) contain a sterol-binding VASt domain, allowing them to facilitate PM to ER cholesterol import[22–24]. Loss of GRAMD1s lead to accumulation of accessible cholesterol in the plasma membrane[22], and mouse macrophages lacking GRAMD1A and GRAMD1B also displayed upreguslated expression of Sterol regulatory-element binding protein 2 (SREBP2) target genes, indicative of decreased ER cholesterol[24]. Due to a limited number of studies on the GRAMD family proteins, their biological importance is still not fully understood. A recent study reported that GRAMD1A activity is required for autophagy initiation[25], suggesting that regulation of cholesterol transport is important to control autophagy levels. GRAMD family proteins were shown to form heterocomplexes in a manner that is dependent on their C-terminal amphipathic helix region[22], indicating that other GRAMD proteins may also regulate autophagy.

Here, we show that cholesterol and the ER-anchored cholesterol transport protein GRAMD1C negatively regulates autophagy initiation. GRAMD1C associates with mitochondria through its GRAM domain and its depletion leads to the accumulation of mitochondrial cholesterol and increased mitochondrial respiration. In addition, we show that members of the GRAMD1 family of cholesterol transport proteins are involved in clear cell renal carcinoma (ccRCC) survival. The GRAMD1s therefore represent potential regulators of the autophagy pathway and ccRCC.

## Results

### Cholesterol depletion promotes autophagy initiation

Previous in vivo and in vitro studies have indicated that cholesterol depletion promotes autophagy[14,26–28], but most of these studies involve cells being depleted of cholesterol for extended time periods, where changes in autophagy can be caused by metabolic and transcriptional responses, thus indirectly activating autophagy. To investigate a more direct role of cholesterol on membrane remodeling during autophagy, we analyzed the short-term effects on starvation-induced autophagy after cholesterol depletion using MBCD, which rapidly removes cholesterol from cellular membranes[29]. U2OS cells were treated with MBCD for 1 h in control (DMEM) or starvation (EBSS) medium in the presence or absence of the lysosomal V-ATPase inhibitor Bafilomycin A1 (BafA1), thus blocking autophagic turnover[30], followed by immunoblotting for the autophagosome markers LC3B and p62, to determine autophagic flux. Cholesterol depletion caused a threefold increase in the membrane-bound form of LC3B (LC3B-II) under basal conditions compared to control cells, which was further enhanced in starved cells depleted of cholesterol (Fig. 1a, b). In line with this, cholesterol depletion increased the formation of autophagosomes both in

fed and starved cells, as assessed by immunofluorescence microscopy of endogenous LC3B puncta (Fig. 1c, d). In both cases, the starvation-induced autophagic flux was higher in MBCD-treated cells, suggesting that cholesterol and amino acid depletion can modulate autophagy synergistically. Similarly, long-term cholesterol depletion using atorvastatin (ATV) for 48 h, an HMG-CoA reductase inhibitor, caused a similar increase in starvation-induced autophagy (Fig. 1e, f), while p62 turnover was not significantly altered with MBCD or ATV treatment (Supplementary Fig. 1a, b).

Autophagosome biogenesis is regulated by the autophagy initiation machinery comprising the ULK1 complex (ATG13, ULK1, FIP200, ATG101), PI3CK3 complex (ATG14L, BECN1, PIK3C3, PIK3R4) and the ATG12-ATG5-ATG16L1 complex. Given that LC3B lipidation was further increased in cells co-treated with MBCD and BafA1 compared to MBCD-treated cells (Fig. 1a–d), we suspected that the MBCD-induced increase in LC3B lipidation was caused by changes in autophagosome biogenesis. In line with this, we observed a significant increase in the number of ATG13 and ATG16L1 positive puncta in U2OS cells treated with MBCD for 1 h, which was further increased upon amino acid starvation (Fig. 1g–i). In support of this, ATV-treated cells also exhibited enhanced ATG13 recruitment at both basal and starved states (Fig. 1j, k). Taken together, our results show that cholesterol depletion promotes autophagy induction and enhances starvation-induced autophagy.

### Cholesterol depletion alters starvation-induced autophagy dynamics

mTORC1 is inactivated when amino acids are not available leading to downstream autophagy induction[31]. Recent reports also indicate that mTORC1 signaling is regulated by lysosomal cholesterol levels, and that lysosomal cholesterol depletion leads to the inactivation of mTORC1[32,33]. To study a possible role of mTORC1 in cholesterol depletion-induced autophagy, we investigated the short-term temporal dynamics of mTORC1 signaling, and autophagic flux in U2OS cells treated or not with either MBCD or ATV and starved at different time points (15, 30, 45, or 60 min) prior to immunoblotting for LC3B and the mTORC1 substrate p70S6K. Surprisingly, starvation-induced LC3B flux was significantly increased at 30 min in starved cells treated with MBCD, while this required 45 min in starved control cells (Fig. 2a, b), indicating a more rapid induction of autophagosome biogenesis in cholesterol-depleted cells. Similar results were observed in ATV-treated cells (Fig. 2c, d). Interestingly, while the mTORC1-specific phosphorylation of p70S6K was completely lost after 15 min of amino acid starvation, this took more than 30 min in cells treated with MBCD (Fig. 2e), indicating a delayed kinetic of mTORC1 inactivation in cells subjected to cholesterol depletion compared to that seen in starved cells. mTORC1 inactivation results in the activation of the ULK1 complex[34]. Intriguingly, we found that ATG13 positive structures formed already after 15 min of MBCD treatment (Fig. 2f, g), before the complete inactivation of mTORC1 (Fig. 2e), suggesting mTORC1-independent activation of ULK1 upon cholesterol depletion.

As cholesterol is often found between the carbon chains of membrane phospholipids, leading to decreased membrane fluidity, we hypothesized that cholesterol removal promotes the generation of curved membranes during de novo synthesis of autophagic membranes. To study this, we generated a membrane curvature reporter based on the amphipathic helix of the BATS domain of ATG14L1, known to bind to curved PtdIns(3)P-enriched membranes destined for autophagosome formation[35] (Supplementary Fig. 1c). U2OS cells stably expressing EGFP-BATS formed puncta when starved for 1 h, where most puncta co-localized with autophagy initiation proteins (ATG13, ATG16L1, and WIPI2), as well as LC3B positive structures (Supplementary Fig. 1d), indicating that it marks early autophagy membranes. As predicted, MBCD treatment caused a significant increase of EGFP-BATS puncta compared to control cells (Fig. 2h, i), indicating that cholesterol depletion induces the formation of curved early autophagy membranes.

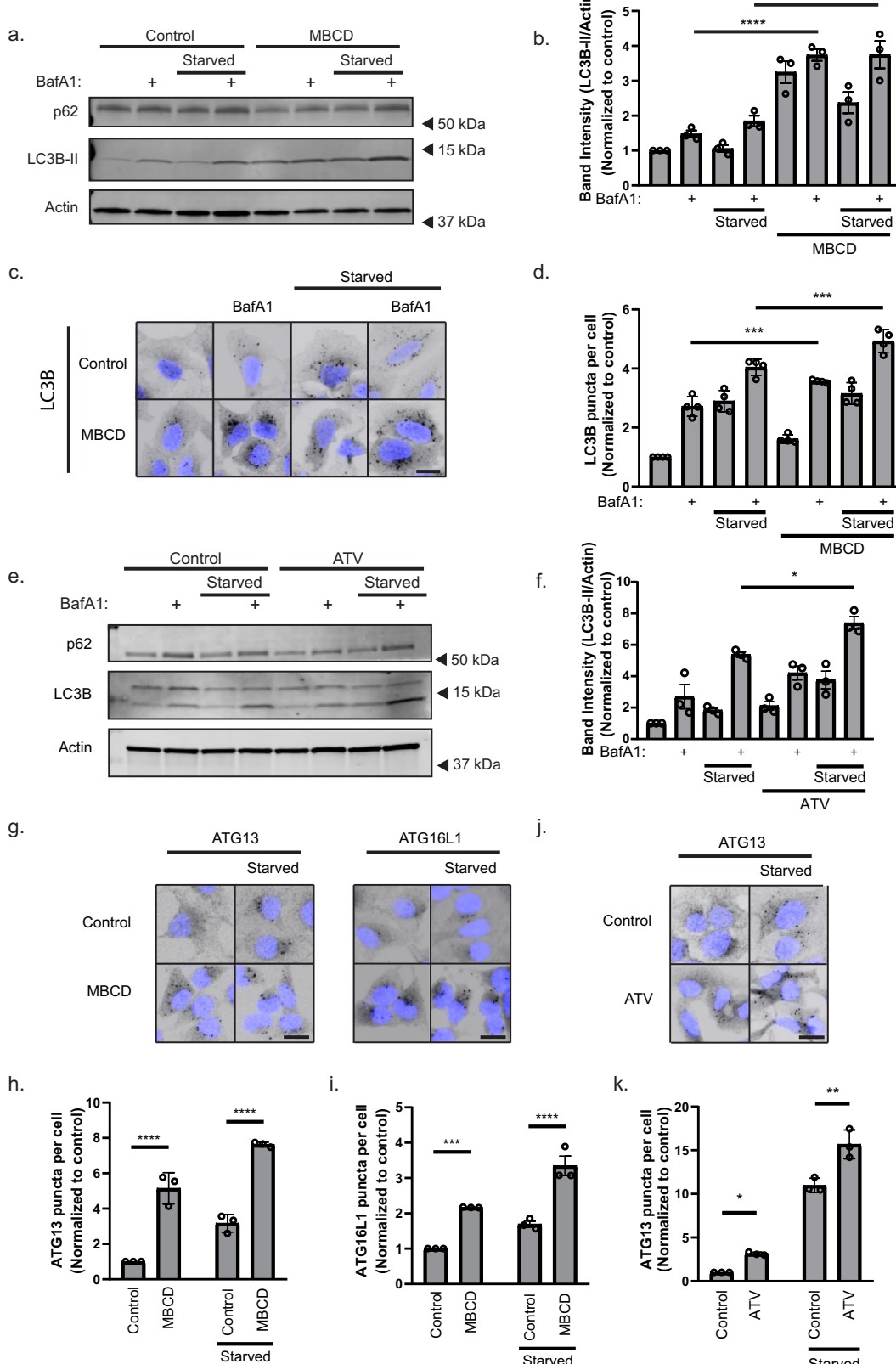

## GRAMD1C is a negative regulator of starvation-induced autophagy

To elucidate whether genetic manipulation of cellular cholesterol regulators also affects autophagy, we decided to deplete cholesterol transport proteins that are mediators of non-vesicular inter-organellar cholesterol movement at membrane contact sites[21]. The GRAMD family proteins, GRAMD1A, GRAMD1B, GRAMD1C, GRAMD2, and GRAMD3 (encoded by *GRAMD1A*, *GRAMD1B*, *GRAMD1C*, *GRAMD2a*, and *GRAMD2b*) are ER-anchored transmembrane proteins[23], known to mediate plasma membrane to ER cholesterol transport[22]. Given that the GRAMs form a complex through their C-terminal region[22] and since GRAMD1A has been implicated in autophagy[25], we asked whether the

**Fig. 1 | Cholesterol depletion promotes autophagy initiation. a** U2OS cells were treated with 2.5 mM methyl-β cyclodextrin (MBCD) in DMEM or in EBSS supplemented ± 10 nM Bafilomycin A1 (BafA1) for 1 h. **b** Quantification of LC3B-II band intensity relative to actin and normalized to DMEM control. Error bars = SEM. Significance was determined using one-way ANOVA followed by Tukey's comparisons test from n = 3 independent experiments. P value <0.001 and 0.004. **c** U2OS cells were treated with 2.5 mM MBCD in DMEM or in EBSS ± 10 nM BafA1 for 1 h. Scale bar = 20 μm. **d** Quantification of LC3B puncta number per cell normalized to the DMEM control. Error bars = SEM. Significance was determined using one-way ANOVA followed by Tukey's comparisons test from n = 4 independent experiments, >500 cells per condition. P value = 0.001 and 0.006. **e** U2OS cells were treated with 10 μM atorvastatin (ATV) for 48 h, before amino acid starvation in EBSS 1 h ±10 nM BafA1. **f** Quantification of LC3B-II band intensity relative to actin and normalized to

DMEM control. Error bars = SEM. Significance was determined using one-way ANOVA followed by Tukey's comparisons test from n = 3 independent experiments. P value = 0.0123. **g** U2OS cells were treated with 2.5 mM MBCD in DMEM or EBSS. Scale bar = 20 μm. **h, i** ATG13 (**h**) and ATG16L1 (**i**) puncta number per cell were normalized to the DMEM control. Error bars = SEM. Significance was determined using two-way ANOVA followed by Sidak's comparisons test from n = 3 independent experiments, >500 cells per condition. P value (ATG13) < 0.0001, <0.0001 and P value (ATG16L1) = 0.0008 and P value <0.0001. **j** U2OS cells were treated with 10 μM ATV for 48 h, before amino acid starvation in EBSS 1 h Scale bar = 20 μm. **k** ATG13 puncta number per cell were normalized to the DMEM control. Error bars = SEM. Significance was determined using two-way ANOVA followed by Sidak's comparisons test from n = 3 independent experiments, >500 cells per condition. P value = 0.0437 and 0.0005. Source data are provided as a Source Data file.

other members of the GRAM family also have a role in autophagy. To study this, U2OS cells with stable inducible expression of the autophagy reporter mCherry-EGFP-LC3B (yellow-autophagosome, red-autolysosome due to quenching of EGFP in the acidic lysosome) were transfected with siRNA to individually deplete each GRAM family member (Supplementary Fig. 1e), followed by quantification of the number of red-only puncta in starved and non-starved cells in the absence or presence of BafA1. Interestingly, there was a significant increase in the number of red-only puncta in GRAMD1C-depleted cells compared to the control, indicating that GRAMD1C is a negative regulator of starvation-induced autophagic flux (Fig. 3a, b). The addition of BafA1 abolished the formation of red-only structures in both control and starved condition, confirming that these structures represent autolysosomes. In line with previous reports[25], 72 h of GRAMD1A depletion promoted basal autophagy flux (Fig. 3a, b). The role for GRAMD1C as a negative regulator of starvation-induced autophagy was further validated in cells depleted of GRAMD1C using two different siRNA oligos (Supplementary Fig. 1f), by measuring the turnover of radioactively labeled long-lived proteins, being predominantly degraded through autophagy[36] and by western blot analysis of starvation-induced turnover of LC3B-II. Indeed, both the starvation-induced turnover of radioactively labeled long-lived proteins (Fig. 3c) and LC3B-II flux (Fig. 3d, e) were increased in GRAMD1C-depleted cells.

To corroborate these findings, we generated GRAMD1C knockout cells (GKO) (Supplementary Fig. 1g). Similar to siRNA-mediated depletion of GRAMD1C, GKO cells exhibited increased LC3B lipidation upon amino acid starvation as compared to their passage-matched wild-type (Wt) counterpart (Fig. 3f, g). Importantly, GKO cells rescued with GRAMD1C (Supplementary Fig. 1h) exhibited lower LC3B-II levels, while GKO cells rescued with GRAMD1C mutants lacking the VASt domain (ΔVASt) (Δ326-497) (Fig. 3f, g) or the GRAM domain (ΔGRAM) (Δ1-205) (Supplementary Fig. 2a, b) did not reduce LC3B-II formation upon starvation. GRAMD1C or ΔGRAM overexpression in Wt cells did not significantly alter LC3 lipidation and turnover (Supplementary Fig. 2c, d). Taken together, our data show that GRAMD1C-mediated cholesterol transport regulates starvation-induced autophagy.

## GRAMD1C is dispensable for mitophagy
To investigate whether GRAMD1C or any of the other GRAMD family proteins regulate selective autophagy, U2OS cells with stable inducible expression of a mitophagy reporter construct (Mitochondria localization signal (MLS)-EGFP-mCherry)[37] and stable ectopic Parkin expression were transfected with siRNA targeting each GRAM family protein, followed by induction of mitophagy by deferiprone (DFP) or CCCP. DFP is an iron chelator that induces a HIF1α-dependent response, leading to induction of Parkin-independent mitophagy[38], while CCCP treatment results in loss of mitochondrial membrane potential and induction of Parkin-dependent mitophagy[39]. Cells were subjected to high-content microscopy and quantification of red-only puncta, as a read-out for mitophagy. Knockdown of

GRAMD1C neither affected Parkin-dependent (Supplementary Fig. 3a, b) nor Parkin-independent mitophagy (Supplementary Fig. 3c, d), while GRAMD1A depletion increased DFP-induced mitophagy (Supplementary Fig. 3c, d). Thus, GRAMD1C is a negative regulator of starvation-induced autophagy that is dispensable for selective clearance of the mitochondria.

## GRAMD1C regulates autophagy initiation
As we found cholesterol depletion to increase autophagy initiation (Figs. 1 and 2), we investigated if depletion of GRAMD1C caused a similar phenotype. Indeed, GRAMD1C depletion in U2OS cells resulted in increased membrane recruitment of several early autophagy machinery components, such as ATG13, ATG16L1 and the PtdIns(3)P effector protein WIPI2b[40], as analyzed by quantification of the respective puncta from control and starved cells (Fig. 4a–d). The increase in ATG13 and ATG16L1 puncta formation was corroborated in GKO cells as compared to passage-matched Wt cells, both before and after amino acid starvation (Fig. 4e–g). ATG16L1 and ATG13 positive structures were found to colocalize with GRAMD1C positive ER sites (Supplementary Fig. 2e), but neither ATG16L1 nor ATG13 co-immunoprecipitated with GRAMD1C (Supplementary Data 1). In line with our results from cholesterol-depleted cells, we found that GRAMD1C depletion promoted the recruitment of EGFP-BATS in starved cells (Fig. 4h, i), suggesting that GRAMD1C inhibits autophagosome initiation by reducing membrane curvature.

Interestingly, we found GRAMD1C itself to be delivered to lysosomes upon starvation, as LAMP1-positive red-only puncta were seen in starved cells expressing mCherry-EGFP-GRAMD1C, which were sensitive to BafA1 (Supplementary Fig. 2f, g). Taken together, our results indicate that GRAMD1C regulates autophagosome biogenesis through the suppression of membrane curvature and the recruitment of early autophagic markers to ER-associated initiation sites.

## The GRAM domain of GRAMD1C mediates its interaction to the mitochondria
GRAMD1C has previously been reported to facilitate cholesterol import from the plasma membrane[23,41], however the yeast GRAM orthologue, Lam6, has been found to be enriched at different organellar contact sites, such as the vCLAMP (vacuole and mitochondria patch), NVJ (nuclear vacuolar junction) and ERMES (ER–mitochondria encounter structures)[42]. Sites of autophagosome biogenesis has been found to overlap with ER–mitochondrial contact sites[43] and we therefore asked if GRAMD1C is also recruited to ER–mitochondrial contact sites. The lipid-binding GRAM domain is thought to be responsible for protein targeting to a specific organelle[23]. Given the importance of the GRAM domain of GRAMD1C for its function in autophagy (Supplementary Fig. 2a, b) and the relative conservation of its GRAM domain to that of Lam6 (37.88% sequence identity) (Supplementary Fig. 4a), we analyzed the localization of stably expressed EGFP-tagged GRAMD1C or GRAM domain only (Fig. 5a) by live cell microscopy. As expected, GRAMD1C-EGFP localized to the ER

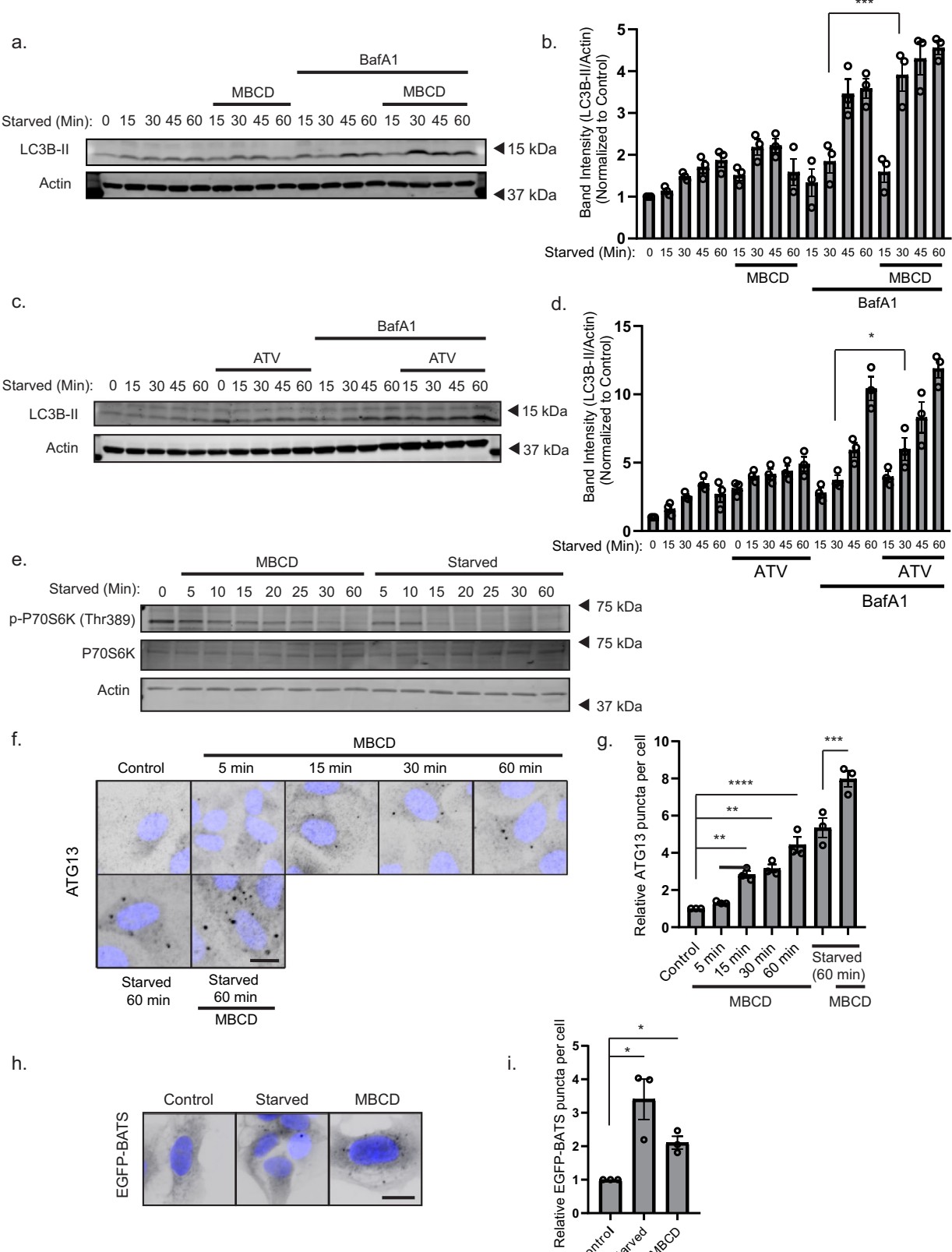

(Supplementary Fig. 4b) and was also found to be enriched at regions of ER–mitochondria overlap (Fig. 5b). Interestingly, the EGFP-GRAM domain localized to structures that associated with mitochondria for a few seconds before dissociating, suggesting that the GRAM domain of GRAMD1C interacts transiently with mitochondria (Fig. 5c). Indeed, the mitochondrial interaction of the GRAM domain was validated by

isolation and immunoblotting of mitochondria from cells expressing EGFP-GRAM, showing the EGFP-GRAM domain both in the crude and pure mitochondrial fractions (Fig. 5d). In an attempt to identify proteins interacting specifically with the GRAM domain of GRAMD1C, and by extension characterize the interactome of GRAMD1C, we carried out co-immunopurification coupled mass-spectrometry (coIP-MS)

**Fig. 2 | Cholesterol depletion alters starvation-induced autophagy dynamics.**
**a** U2OS cells were starved in EBSS and treated or not with 2.5 mM methyl-β cyclodextrin (MBCD) in DMEM ± 100 nM Bafilomycin A1 (BafA1) for the indicated times before being lysed and subjected to western blot analysis. **b** Quantification of LC3B-II band intensity relative to Actin and normalized to DMEM control. Error bars = SEM. Significance was determined using two-way ANOVA followed by Sidak's comparisons test from $n = 3$ independent experiments. $P$ value = 0.0002. **c** U2OS cells were treated or not with 10 μM atorvastatin (ATV) for 48 h before starvation in EBSS ± BafA1 for the indicated times. **d** Quantification of data in (**c**). as LC3B-II band intensity relative to Actin and normalized to DMEM control. Error bars = SEM. Significance was determined using two-way ANOVA followed by Sidak's comparisons test from $n = 3$ independent experiments. $P$ value = 0.0454. **e** U2OS cells were treated for the indicated times with either 2.5 mM MBCD in DMEM or EBSS without MBCD before western blot analysis with the indicated antibodies. **f** U2OS cells were treated with 2.5 mM MBCD in DMEM or EBSS for the indicated time before fixation, immunostaining for ATG13, and wide-field microscopy. Scale bar = 20 μm. **g** Quantification of ATG13 puncta per cell normalized to the control. Error bars = SEM. Significance was determined using one-way ANOVA followed by Dunnett's comparisons test from $n = 3$ independent experiments, >500 cells per condition. $P$ value = 0.0151, 0.0042, <0.0001, 0.0007. **h** Cells expressing EGFP-BATS were incubated in EBSS or serum-free DMEM supplemented or not with 2.5 mM MBCD for 1 h before fixation and wide-field microscopy. Scale bar = 20 μm. **i** Quantification of EGFP-BATS puncta in (**h**). Error bar = SEM. The number of EGFP-BATS puncta per cell was normalized to the DMEM control. Significance was determined using one-way ANOVA followed by Dunnett's post test from $n = 3$ independent experiments, >500 cells per experiment. $P$ value = 0.0166 and 0.0024. Source data are provided as a Source Data file.

analysis of the interactome of EGFP-tagged GRAMD1C and GRAMD1C (ΔGRAM). As expected, GRAMD1A, GRAMD1B, and GRAMD3 were detected as the main interactors of GRAMD1C[22], while GRAMD2 was absent (Fig. 5e and Supplementary Data 1). Interestingly, several mitochondrial proteins were detected in the GRAMD1C interactome, amongst them the outer mitochondrial membrane protein TOMM70A (Fig. 5e and Supplementary Data 1). Given that Lam6 interacts with mitochondria via binding to Tom70[42], we suspect a similar mechanism of mitochondrial recruitment for GRAMD1C. To further investigate the mechanism responsible for the association of GRAMD1C to the mitochondria, we carried out western blot analysis of immunoprecipitates of EGFP-GRAM, GRAM1C-EGFP, GRAMD1C (ΔGRAM)-EGFP. All proteins were able to pull down the outer mitochondrial membrane protein RHOT2 (also known as MIRO2) and the ER–Mitochondrial contact site marker ACSL4 (Fig. 5f), confirming the interaction of the GRAM domain with the mitochondria. We suspect that the mitochondrial interaction of GRAMD1C (ΔGRAM)-EGFP is mediated through its interaction with other GRAM family proteins, such as GRAMD1A and GRAMD1B, as the GRAM domain only was unable to bind these proteins (Fig. 5f). The EGFP-GRAM domain was successful in pulling down TOMM70A (Fig. 5f), indicating that the interaction to mitochondria is facilitated by the GRAM domain. Interestingly, cholesterol synthesis proteins such as SQLE, DHCR24, and DHCR7 were identified in the GRAMD1C immunoprecipitate (Fig. 5e and Supplementary Data 1), suggesting that the GRAM complex is localized in proximity to regions of cholesterol synthesis. As the interaction between mitochondria and ER can be affected by changes in mitochondrial structure, we analyzed mitochondrial morphology upon depletion of the different GRAMs, but did not see any changes to mitochondrial morphology (Supplementary Fig. 4c, d). Depletion of GRAMD1C did also not affect mitochondrial width (Supplementary Fig. 4e). Taken together, our results showing that GRAMD1C interacts with mitochondrial proteins, ER–mitochondria contact site proteins and proteins involved in cholesterol synthesis, indicate a role in ER–mitochondria cholesterol transport.

## GRAMD1C regulates mitochondrial bioenergetics

To investigate whether GRAMD1C can potentially regulate cholesterol movement between the ER and the mitochondria, we developed a method for mitochondrial cholesterol quantification based on the addition of recombinant mCherry-tagged cholesterol-binding domain of Perfringolysin O (mCherry-D4)[44] to isolated mitochondria. As expected, MBCD treatment decreased mCherry-D4 binding to purified mitochondria, indicating that mCherry-D4 selectively binds to cholesterol on isolated mitochondria (Supplementary Fig. 4f). Importantly, the levels of mCherry-D4 bound to mitochondria isolated from GRAMD1C KO cells were increased compared to control cells (Fig. 6a, b), indicating that GRAMD1C regulates mitochondrial cholesterol levels. In support of this, cholesterol oxidase-based quantification of mitochondrial cholesterol revealed a similar increase of mitochondrial

cholesterol in GRAMD1C KO cells compared to control cells (Fig. 6c). Electron microscopy analysis showed that mitochondria of siGRAMD1C-treated cells are weakly stained by OsO4 (an unsaturated fatty acid stain), suggesting altered mitochondrial lipid composition (Fig. 6d). Moreover, the expression of several SREBP2 target genes was increased in GRAMD1C-depleted cells (Supplementary Fig. 4g), suggesting that loss of GRAMD1C caused a reduction of ER cholesterol levels, in line with a previous observations[22]. Cell fractionation further revealed that GRAMD1C can be found in crude mitochondria supporting a role for GRAMD1C at the ER–mitochondria interface (Supplementary Fig. 4h). Thus, our data indicate that GRAMD1C facilitates cholesterol transport between the mitochondria and the ER.

To investigate whether the increased mitochondrial cholesterol levels seen in GRAMD1C-depleted cells affects the mitochondrial bioenergetics, we used the Seahorse XF Analyzer to measure ATP-production-linked respiration and the maximal respiratory capacity. Indeed, both were significantly increased in cells lacking GRAMD1C (Fig. 6e, f). Increased ATP-linked respiration was also seen in GKO cells, which was reversed upon rescue with GRAMD1C (Fig. 6g–h and Supplementary Fig. 5a, b), but not with GRAMD1C lacking the VASt domain (Supplementary Fig. 5a, b), indicating that mitochondrial cholesterol levels regulate respiration. Surprisingly, GKO cells expressing the ΔGRAM mutant exhibited reduced ATP-linked respiration (Fig. 6g, h), which possibly reflects the ability of the ΔGRAM mutant to interact with mitochondria (Supplementary Fig. 4g). Western blot and mass-spectrometry analysis did not reveal significant changes to the OXPHOS proteins in GRAMD1C knockdown cells (Supplementary Fig. 5c–e and Supplementary Data 2), suggesting that the increased mitochondrial respiration was not caused by changes in the mitochondrial proteome. Notably, the abundance of several cholesterol-associated proteins (STARD9, ERLIN, SQLE, NPC2, and APOB) in GRAMD1C-depleted cells were altered (Supplementary Fig. 5e and Supplementary Data 2), however, the significance of this is not known. Similarly, neither mitochondrial membrane potential nor total cellular reactive oxygen species (ROS) were altered in GRAMD1C knockdown cells (Supplementary Fig. 5f, g). In summary, our results indicate that GRAMD1C is a negative regulator of mitochondrial cholesterol abundance and mitochondrial bioenergetics.

## GRAM family expression is prognostic in ccRCC

GRAM proteins have previously been implicated in tumorigenesis, as GRAMD1B depletion was found to promote breast cancer cell migration[45], while *GRAMD1C* transcript levels seem to positively correlate with the level of immune cell infiltration and overall survival in Clear Cell Renal Carcinoma (ccRCC) patients[46]. ccRCC is a subtype of kidney cancer that stems from the epithelial cells of the proximal convoluted tubule of the kidney[47], and is characterized by altered mitochondrial metabolism and aberrant lipid and cholesterol accumulation[48]. Given that *GRAMD1C* expression correlates with overall survival in ccRCC[46] and forms a heteromeric complex with the

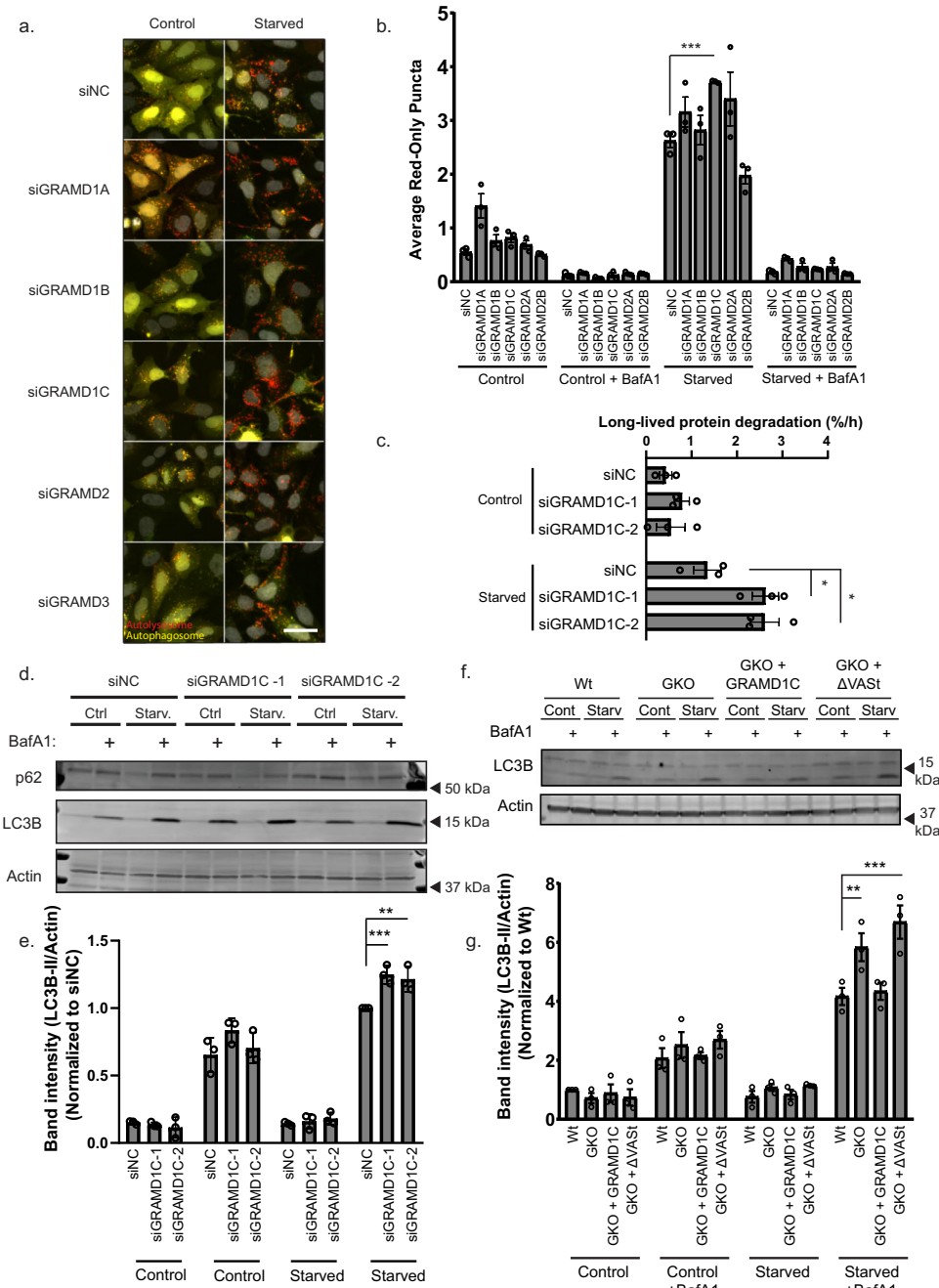

**Fig. 3 | GRAMD1C is a negative regulator of starvation-induced autophagy.**
**a** U2OS cells stably expressing mCherry-EGFP-LC3B were transfected with siRNA against the indicated genes for 72 h before serum and amino acid starvation in EBSS for 2 h ± 100 nM Bafilomycin A1 (BafA1). Scale bar = 20 μm. **b** Quantification of data in (**a**). The bars represent the average number of red-only puncta per cell. Significance was determined using two-way ANOVA followed by Dunnett's comparison test from $n = 3$ independent experiments, >500 cells per condition. Error bar = SEM. $P$ value <0.0001. **c** U2OS cells were incubated in culture media supplemented with [14]C valine for 24 h, then washed and re-incubated in media containing non-radioactive valine for 16 h to allow degradation of short-lived proteins. The cells were then starved ±100 nM BafA1 for 4 h, followed by the analysis of radioactive [14]C valine in the media and cells using a liquid scintillation counter. Significance was determined using a two-way followed by Tukey's comparison test from $n = 3$ independent samples. $P$ value = 0.0133 and 0.0145. **d** U2OS cells were transfected

with control siRNA (siNC) or two different siRNA targeting GRAMD1C for 72 h, followed by incubation in DMEM (Ctrl) or EBSS (Starv.) for 2 h ±100 nM BafA1. **e** Quantification of band intensity of LC3B-II relative to Actin in (**d**). Values are normalized to siNC starved + BafA1. Error bar = SEM. Significance was determined using two-way ANOVA followed by Tukey's comparison test from $n = 3$ independent experiments. $P$ value = 0.0008 and 0.0036. **f** Wild-type (Wt), GRAMD1C knockout cells (GKO) and GKO cells stably expressing GRAMD1C (GKO + GRAMD1C) and GKO cells stably expressing GRAMD1C lacking the VASt domain (GKO + ΔVASt) were incubated in DMEM or EBSS for 2 h ±100 nM BafA1. **g** Quantification of band intensity of LC3B-II relative to Actin in (**f**). Values are normalized to Wt control. Error bar = SEM. Significance was determined using two-way ANOVA followed by Tukey's comparison test from $n = 3$ independent experiments. $P$ value = 0.0011 and <0.0001. Source data are provided as a Source Data file.

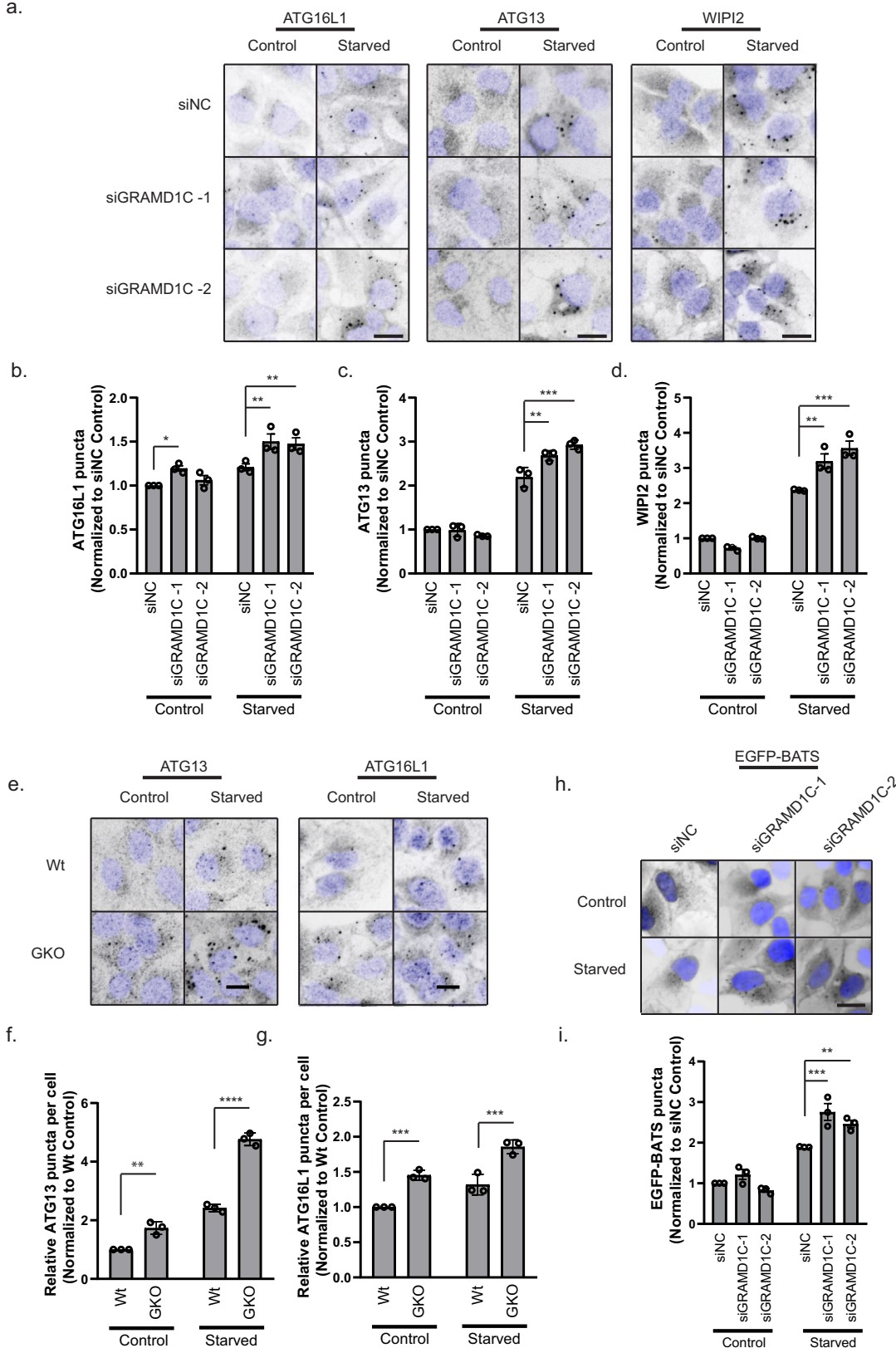

other GRAMs[22], we investigated the involvement of the complete GRAM family in ccRCC using tumor gene expression data from the TCGA Kidney Renal Clear Cell Carcinoma (KIRC) cohort[49]. Interestingly, the expression of several GRAM family members was significantly associated with survival outcome. Similar to *GRAMD1C*, high *GRAMD2B* expression was associated with improved patient survival in

ccRCC. By contrast, low expression of *GRAMD1A* and *GRAMD1B* was favorable with respect to survival (Fig. 7a). We further found a weak negative correlation between *GRAMD1C* expression and *GRAMD1A* and *GRAMD1B* levels (Supplementary Fig. 6a), reflecting their differential influence on overall survival. Furthermore, while *GRAMD1C* expression was decreased in advanced-stage tumors, the expression of *GRAMD1A*

**Fig. 4 | GRAMD1C regulates autophagy initiation during amino acid starvation.** **a** U2OS cells were transfected with siRNA against control (siNC) or GRAMD1C for 72 h prior to starvation in EBSS for 1 h or incubation in DMEM (control). The cells were then fixed, immunostained with antibodies against ATG16L1, ATG13 or WIPI2 and subjected to wide-field microscopy. Scale bar = 20 μm. The number of **b** ATG16L1, **c** ATG13, and **d** WIPI2 puncta per cell were quantified and normalized to siNC control from *n* = 3 independent experiments, >500 cells per condition. Significance was determined using two-way ANOVA followed by Tukey's comparison test. Error bar = SEM. *P* value (ATG16L1) = 0.0441, 0.0042, and 0.008, *P* value (ATG13) = 0.0013 and <0.0001, *P* value (WIPI2) = 0.0014 and <0.0001. **e** Wild-type (Wt) or GRAMD1C knockout (GKO) U2OS cells were starved or not in EBSS for 1 h before immunostaining for ATG16L1 or ATG13. Scale bar = 20 μm. **f, g** The number of ATG13 (**f**) and ATG16L1 (**g**) puncta per cell were quantified and normalized to Wt control from *n* = 3 independent experiments, >500 cells per condition. Significance was determined using two-way ANOVA followed by Sidak's comparison test. *P* value (ATG13) = 0.0012 and <0.0001, and *P* value (ATG16L1) = 0.0008 and 0.0003. **h** U2OS cells expressing EGFP-BATS were transfected with siRNA against control or GRAMD1C for 72 h before starvation in EBSS for 1 h. Scale bar = 20 μm. **i** The number of EGFP-BATS puncta per cell was quantified and normalized to siNC control from *n* = 3 independent experiments, >500 cells per condition. Significance was determined using two-way ANOVA followed by Tukey's comparison test. Error bar = SEM. *P* value = 0.0003 and 0.0069. Source data are provided as a Source Data file.

and *GRAMD1B* showed an opposite pattern, with increased expression in late-stage tumor samples compared to early-stage tumor samples (Supplementary Fig. 6b).

To further substantiate these observations, we did a colony-formation assay in 786-O ccRCC cells transfected with siRNA against all GRAMs, showing that depletion of GRAMD1A and GRAMD1B significantly decreased the ability of 786-O cells to form colonies (Fig. 7b, c), supporting the role of these genes in ccRCC survival. Ki67, a marker of cell proliferation was also significantly reduced in GRAMD1A, GRAMD1B and GRAMD1C-depleted 786-O cells (Fig. 7d, e). As overall survival is also affected by invasive and migration capabilities of the tumor, we analyzed migration of 786-O cells depleted of GRAMD1A, GRAMD1B and GRAMD1C using a wound-healing assay. However, only the migration of siGRAMD1B treated cells was slightly decreased (Supplementary Fig. 7a, b). Similar to our observation in U2OS cells, GRAMD1C depletion promoted starvation-induced autophagy in the kidney cell line A498 (Fig. 7f, g) and ATP-production linked respiration in ccRCC 786-O cells (Fig. 7h), indicating that the relationship between mitochondrial bioenergetics and GRAMD1C is conserved among cell lines.

To better understand the role of GRAMD1C in ccRCC, we analyzed the genes co-expressed with *GRAMD1C*, as co-expressed gene networks can allow the identification of functionally related genes[50]. Interestingly, *GRAMD1C* is co-expressed with several mitochondrial genes in ccRCC samples, including AUH, MICU2, and SIRT5, which all moderately correlated with GRAMD1C with Pearson's correlations values of above 0.45 (Supplementary Fig. 6c). In conclusion, members of the GRAM family contribute to the regulation of overall survival of ccRCC patients, possibly through modulation of metabolism and cancer cell survival.

## Discussion

In this study, we investigated the effects of cholesterol and the cholesterol transport protein GRAMD1C on starvation-induced autophagy. We show that cholesterol depletion promotes autophagy initiation and enhances starvation-induced autophagy flux. Similarly, we show that depletion of the cholesterol transporter GRAMD1C promotes starvation-induced autophagy, but has no effect on Parkin-dependent or -independent mitophagy. Importantly, we find GRAMD1C to interact with mitochondria through its PH-like GRAM domain and show that both the GRAM domain and the cholesterol transport VASt domain are important for GRAMD1C-mediated regulation of autophagosome biogenesis. Depletion of GRAMD1C leads to increased mitochondrial cholesterol abundance and increased mitochondrial bioenergetics. Finally, we identify the GRAM family as genes involved in ccRCC survival, highlighting the pathophysiological relevance of cholesterol transport proteins.

Autophagosomes are small vesicles (~0.5–1 μm) formed de novo from ER-associated sites termed omegasomes[51]. Autophagosome biogenesis involves recruitment of the core autophagy machinery and require a high degree of curvature and flexibility, which must be partially supported by specialized lipid compositions. Reflecting this, localized fatty acid synthesis has been shown to occur at autophagy

biogenesis sites and are required for autophagosome generation[4,52,53]. Furthermore, the autophagy proteins ATG2 and ATG9 have been shown to facilitate lipid delivery to growing autophagosomes[54–56]. The role of cholesterol during autophagosome biogenesis remains to be clarified, but given that membrane cholesterol increases membrane rigidity, it is likely that high cholesterol abundance at autophagosome initiation sites is unfavorable for autophagosome biogenesis.

Here, we show that cholesterol depletion has a synergistic effect on autophagic flux in response to amino acid starvation, suggesting that cholesterol and amino acid depletion activate autophagy in part through mutually exclusive mechanisms. However, as cholesterol levels also regulate mTORC1 signaling[33], we cannot rule out the involvement of mTORC1 signaling in cholesterol depletion-induced autophagy. Our data also indicate that short-term cholesterol depletion leads to a change in the membrane curvature at autophagosome initiation sites, as membrane recruitment of autophagy initiation proteins ATG13, WIPI2, and ATG16L1, and the curvature-sensing BATS domain of ATG14L were significantly increased in cells depleted of cholesterol. As ATG14L is recruited to PtdIns(3) P-enriched ER-associated omegasomes upon induction of autophagy by starvation[57], we speculated that a removal of cholesterol at such sites might facilitate autophagosome biogenesis.

The proteins of the GRAMD family are ER-anchored transmembrane proteins that function as cholesterol transport proteins[22] that previously have been shown to associate with the PM to facilitate ER cholesterol import[23]. We here demonstrate that GRAMD1C depletion promotes starvation-induced autophagy by increasing the number of autophagosome initiation sites. Both the GRAM and the VASt domains of GRAMD1C are important for its role as a negative regulator of starvation-induced autophagy. We show that GRAMD1C interacts with mitochondria through its GRAM domain and co-precipitates with mitochondrial proteins. Most interestingly, mitochondrial cholesterol levels were increased in GRAMD1C knockout cells, suggesting that it regulates cholesterol transport between mitochondria and the ER.

Lam6, the yeast orthologue of GRAMD1C was previously shown to be recruited to mitochondria through its interaction with Tom70[42]. We find that GRAMD1C also associates with TOMM70A and other outer mitochondrial membrane proteins. The GRAM domains of GRAMD1s are cholesterol sensors that bind to different target membranes[58], including the PM and mitochondria. However, GRAMD1C lacking the GRAM domain (ΔGRAM) was able to interact with mitochondrial proteins, most likely due to its ability to form a complex with other GRAMD1s. Alternatively, the GRAMD1C might interact with mitochondria independent of the GRAM domain, similar to GRAMD1B having a mitochondrion-targeting sequence upstream of its GRAM domain[59]. Interestingly, GRAMD1C copurified with ACSL4 and TMX1, both markers for ER–mitochondria contact sites, supporting the localization of GRAMD1C to these regions. In addition, GRAMD1C interacts with cholesterol synthesis proteins (Supplementary Data 1), suggesting that the GRAMDs facilitate cholesterol redistribution after cholesterol synthesis. We were not able to establish if GRAMD1C is a bona fide contact tether[60], or if this interaction is transient in nature. Nevertheless, autophagy initiation has been shown to mainly occur at

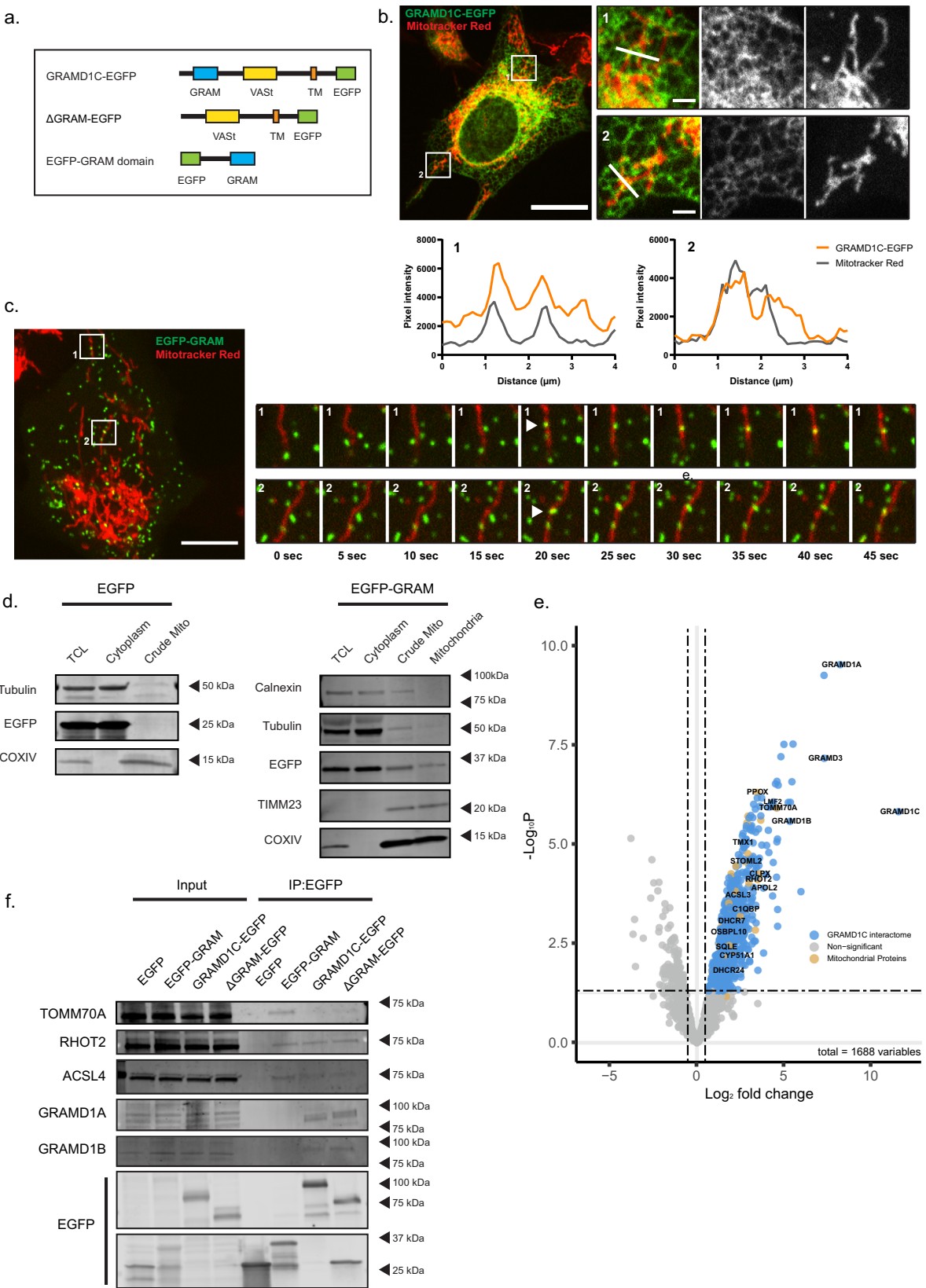

ER–mitochondria contact sites[43,61], thus placing GRAMD1C at sites of autophagosome biogenesis. GRAMD1C localizes to the ER and interacts with mitochondria and the plasma membrane via its GRAM domain to facilitate cholesterol transport from these cellular structures to the ER. Thus, we propose a model where GRAMD1C contributes to the suppression of autophagosome biogenesis by

modulating cholesterol levels at ER membranes that are associated with autophagosome initiation sites (Supplementary Fig. 7c).

Recent reports suggest the involvement of GRAMD1A in the regulation of autophagosome biogenesis[25]. It was reported that while 24 h of GRAMD1A depletion inhibited LC3 puncta formation, this effect was not observed after 72 h knockdown of GRAMD1A[25], suggesting

**Fig. 5 | GRAMD1C interacts with the mitochondria through the GRAM domain.** **a** Overview of the EGFP-tagged GRAMD1C constructs used. TM = Transmembrane domain. **b** U2OS cells stably expressing GRAMD1C-EGFP were stained with Mitotracker Red and subjected to live cell confocal microscopy. The graphs depict the pixel intensity along the white lines drawn in the insets (1 and 2). Scale bar = 20 μm, inset = 2 μm. *n* = 3 independent experiments, >5 cells each experiment. **c** U2OS cells stably expressing EGFP-GRAM domain were stained with Mitotracker Red and subjected to live cell confocal microscopy. The insets (1 and 2) represent snapshots showing the recruitment of EGFP-GRAM to the mitochondria. Scale Bar = 10 μm. *n* = 3 independent experiments, >5 cells each experiment. **d** Mitochondria were isolated using percoll density centrifugation from U2OS cells expressing EGFP or EGFP-GRAM and subjected to western blot analysis for the indicated proteins. TCL total cell lysate. *n* = 3 independent experiments **e** GRAMD1C-EGFP was immuno-purified from stably transfected U2OS cells, and copurified proteins were identified using mass-spectrometry analysis. The interactome of GRAMD1C-EGFP was compared to the interactome of EGFP. Significant hits (*P* < 0.05) are colored blue or brown (mitochondrial). **f** EGFP, EGFP-GRAM, GRAMD1C-EGFP, and ΔGRAM-EGFP were immunoprecipitated from stably transfected U2OS cells and the precipitate subjected to western immunoblot analysis for the indicated proteins. *n* = 3 independent experiments. Source data are provided as a Source Data file.

adaptations to GRAMD1A loss or a complex interplay between the different GRAM proteins. Indeed, 72 h of GRAMD1A depletion in U2OS cells also did not inhibit autophagy as previously reported in MCF7 and HEK293T cells[25], but rather enhanced basal autophagy flux. A recent study found that GRAMD1C knockout immortalized mouse myoblast C2C12 cells exhibit increased LC3-II, likely by regulating lysosomal trafficking and mTORC1 activity[62]. We find that the effect of GRAMD1C depletion on autophagy is less drastic as compared to MBCD-mediated cholesterol depletion, indicating that GRAMD1A and GRAMD1B partially compensate for the loss of GRAMD1C given their highly similar structures. However, GKO cells exhibited a more pronounced induction of autophagy initiation events, indicating that GRAMD1C is not completely redundant in the long-term. In addition to mitochondrial proteins, we find GRAMD1C to interact with proteins from different organelles (Supplementary Data 1) and given previous reports of the PM[63], endosomes and Golgi apparatus[64] in autophagosome biogenesis, it is possible that cholesterol transport of other organelles than the mitochondria contributes to the autophagy phenotype observed in GRAMD1C-depleted cells. Furthermore, GRAMD1C did neither affect Parkin-dependent nor -independent mitophagy, possibly reflecting a difference in the de novo formation of the autophagosome during selective and non-selective autophagy.

The increased mitochondrial cholesterol levels seen in GRAMD1C-depleted cells is reminiscent of the increased mitochondrial cholesterol found in Niemann Pick C1 (NPC1) depleted cells[65]. However, the relationship between mitochondrial cholesterol and respiration is not clear. Previous studies have found that mice fed with a cholesterol-enriched diet displayed increased mitochondrial cholesterol and decreased mitochondrial respiration[66], but cholesterol removal with MBCD[67] and simvastatin have also been reported to decrease mitochondrial respiration[68]. This discrepancy can possibly be attributed to the different experimental models and cholesterol loading/depletion systems used. In addition, mitochondria in siGRAMD1C-treated cells are poorly stained with OsO4 (Fig. 6d), a lipid stain that preferably binds to unsaturated fatty acids and can be affected by cholesterol[69], suggesting altered mitochondrial lipid composition in cells lacking GRAMD1C. Our results suggest that mitochondrial cholesterol accumulation caused by the loss of GRAMD1C promotes oxidative phosphorylation. While we have not been able to establish a mechanism for this, proteomic analysis of GRAMD1C-depleted cells suggests an altered composition of proteins involved in cellular metabolism. The abundance of glycogen synthase kinase 3 beta (GSK3B), glycogenin-1 (GYG1) and glycerol-3-phosphate phosphatase (PGP), proteins involved in glycogen synthesis, were decreased in GRAMD1C-depleted cells. In contrast the glycolysis regulator 6-phosphofructo-2-kinase/Fructose-2,6-bisphosphatase 4 (PFKFB4) was increased in cells treated with siGRAMD1C (Supplementary Fig. 5c). The significance of these changes is not clear, but it suggests that metabolic rewiring accompanies the loss of GRAMD1C.

ccRCC cells exhibit a disrupted cholesterol homeostasis, accumulating up to eight times more cholesterol compared to normal kidney cells[70,71]. However, this increase does not appear to stem from increased cholesterol synthesis and possibly originates from aberrant cholesterol transport and metabolism[70]. It is therefore interesting that high expression of GRAMD1C and GRAMD2B was found to associate with improved survival of ccRCC patients, while the opposite was found for GRAMD1A and GRAMD1B. This suggests that the GRAMDs may have opposing roles in ccRCC carcinogenesis and survival despite their domain similarities. Critically, mirroring the observation on overall survival, we found that the depletion of *GRAMD1A* and *GRAMD1B* in 786-O ccRCC cells caused a significant decrease in cell survival and proliferation. Moreover, GRAMD1C was found to regulate autophagy and mitochondrial bioenergetics. However, the correlation between GRAMD1 expression levels and patient overall survival is likely more complex than just cell proliferation, changes to mitochondrial respiration or autophagy. As GRAMD1C is a regulator of autophagy in ccRCC, it will be interesting to investigate fluid flow[72] and sheer stress[73] induced autophagy in the kidney and renal carcinoma. Given the importance of autophagy in ccRCC cell growth[74] and therapeutic resistance[75] the GRAMDs represent relevant therapeutic targets for ccRCC.

In conclusion, our results show that short-term cholesterol depletion is favorable for autophagosome biogenesis by increasing the membrane recruitment of early core autophagy proteins. We show that depletion of the ER-anchored cholesterol transport protein GRAMD1C promotes starvation-induced autophagy and demonstrate that GRAMD1C interacts with mitochondria to facilitate mitochondria-ER cholesterol transport. Finally, we find that the expression of various *GRAM* genes correlates with ccRCC survival. These results underline the importance of cholesterol transport proteins in autophagy and mitochondrial bioenergetics and warrants further investigation into the regulation of membrane cholesterol during autophagosome biogenesis and cancer.

## Methods
### Antibodies
The following primary antibodies were used: anti-LC3B (#3868, Cell Signaling, Western blotting (WB), 1:1000), anti-LC3B (#PM036, MBL, immunofluorescence microscopy (IF), 1:500), anti-p62 (#610833, BD biosciences, WB, 1:1000), anti-GRAMD1C (#HPA012316, Sigma, WB, 1:500), anti-TOMM70A (#SAB1401493, Sigma, WB, 1:1000), anti-tubulin (#T5168, Sigma, WB, 1:5000), anti-actin (#3700, Cell Signaling, WB, 1:5000), anti-EGFP (#632381, Takara, WB, 1:1000), anti-mCherry (#PA534974, Thermo Fisher, WB, 1:1000), anti-TOM20 (#17764, Santa Cruz, WB, 1:1000), anti-ACSL4 (#sc-365230, Santa Cruz, WB, 1:1000), anti-TIM23 (#611223, BD Biosciences, WB, 1:1000), anti-COX IV (#4850, Cell Signaling, WB, 1:1000), anti-ATG13 (#13468, Cell Signaling, IF, 1:500), anti-ATG16L1 (#PM040, MBL, IF, 1:500), anti-WIPI2 (#Ab105459, Abcam, IF 1:500), total OXPHOS antibody (#ab110413, Abcam, WB, 1:500), anti-PDH (#2784S, Cell Signaling, WB 1:1000), anti-GAPDH (#5174, Cell Signaling, WB 1:5000), p70S6K (#9202, Cell Signaling, WB, 1:1000), phospho-P70S6K (Thr389) (#9205, Cell Signaling, WB, 1:1000), anti-MIRO2 (#PA5-52960, Proteintech, WB, 1:1000).

Secondary antibodies for western blotting used were anti-mouse DyLight 680 (#SA5-10170, Thermo Fisher, 1:10,000), anti-rabbit DyLight 800 (#SA5-10044, Thermo Fisher, 1:10,000). Secondary antibodies used for immunofluorescence were Anti-rabbit Alexa Fluor 488

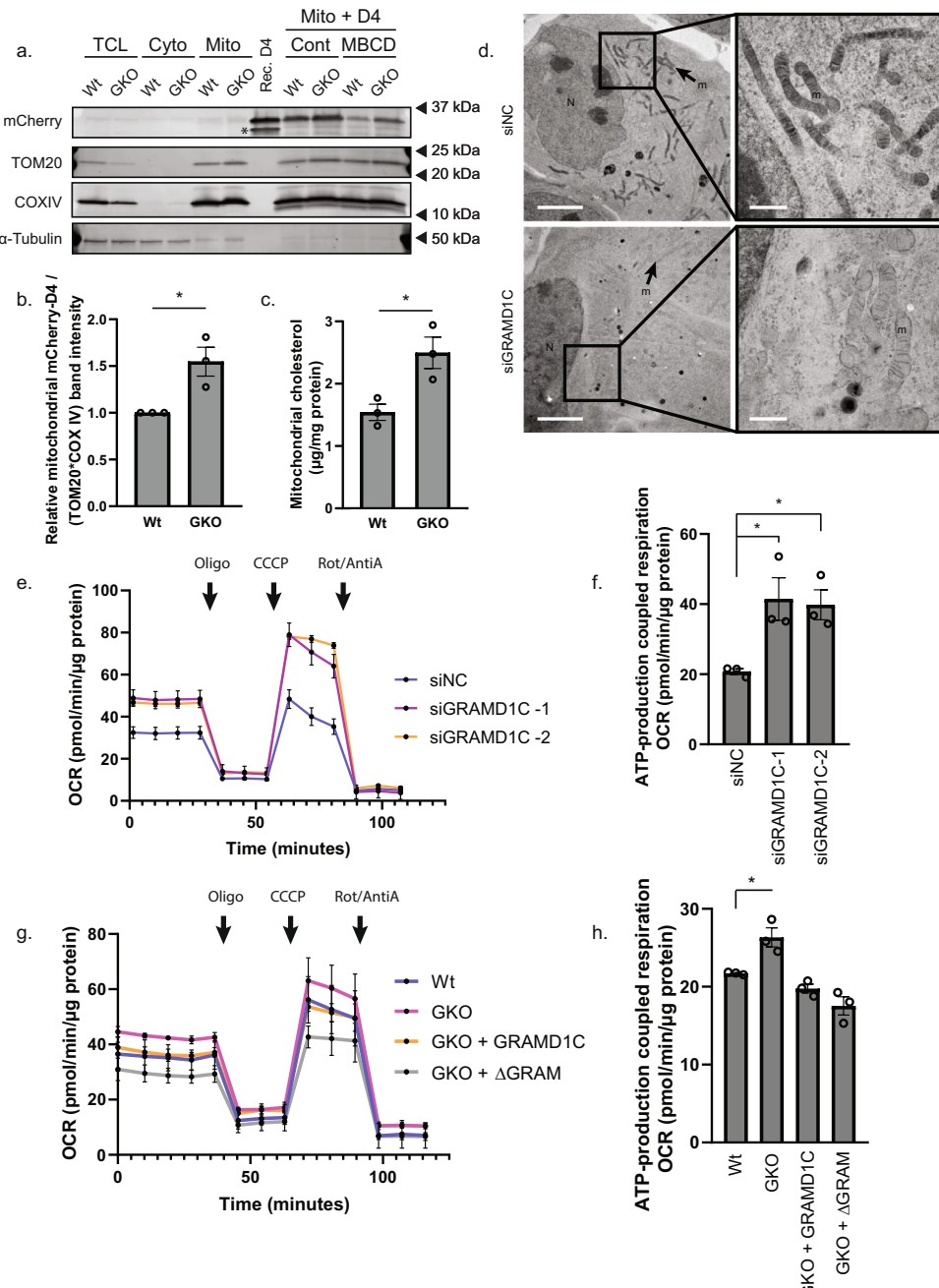

**Fig. 6 | GRAMD1C regulates mitochondrial bioenergetics. a** Wt and GRAMD1C knockout (GKO) cells expressing 3XHA-EGFP-OMP25 were subjected to mitochondria isolation from a total cell lysate (TCL). The isolated mitochondria were incubated with the cholesterol probe mCherry-D4 (recombinant protein shown in lane 7) in the presence or absence of methyl-β cyclodextrin (MBCD). The asterisk indicates free mCherry. **b** Quantification of mCherry-D4 band intensity in the isolated mitochondria fractions from a. relative to the average band intensity of the mitochondrial proteins TOM20 and COXIV and normalized to Wt cells. Significance was determined using two-tailed Students T test from $n = 3$ independent experiments. Error bar = SEM. $P$ value = 0.0238. **c** Mitochondria from Wt or GKO cells were isolated and mitochondrial lipids were extracted. Absorbance changes corresponding to cholesterol abundance were normalized to mitochondrial protein concentration. Significance was determined using Student's $T$ test from $n = 3$ independent experiments. Error bar = SEM. $P$ value = 0.0162. **d** Cells treated with control siRNA (siNC) or siGRAMD1C were treated to high-pressure freezing followed by freeze substitution (1% OsO4, 0.5% uranyl acetate, 0.25% glutaraldehyde in

acetone). Arrows with m point towards mitochondria. Scale bar = 5 μm, inset scale bar = 1 μm. $n = 31$ cells. **e** Mitochondrial oxygen consumption rate (OCR) was analyzed in control, and GRAMD1C knocked down cells using the Seahorse analyzer. OCR was measured after the gradual addition of Oligomycin (oligo), carbonyl cyanide m-chlorophenyl hydrazone (CCCP) and Rotenone/Antimycin A (Rot/AntiA). **f** ATP-linked respiration is calculated from the difference between the maximal respiratory capacity and the proton leak. Significance was determined using one-way ANOVA followed by Bonferroni's comparison test from $n = 3$ experiments. Error bar = SEM. $P$ value = 0.0294 and 0.0413. **g** OCR was analyzed in Wt, GKO, GKO + GRAMD1C, and GKO + ΔGRAM cells using the Seahorse analyzer. **h** ATP-linked respiration is calculated from the difference between the maximal respiratory capacity and the proton leak. Significance was determined using one-way ANOVA followed by Dunnett's multiple comparison test from $n = 3$ experiments. Error bar = SEM. $P$ value = 0.0159. Source data are provided as a Source Data file.

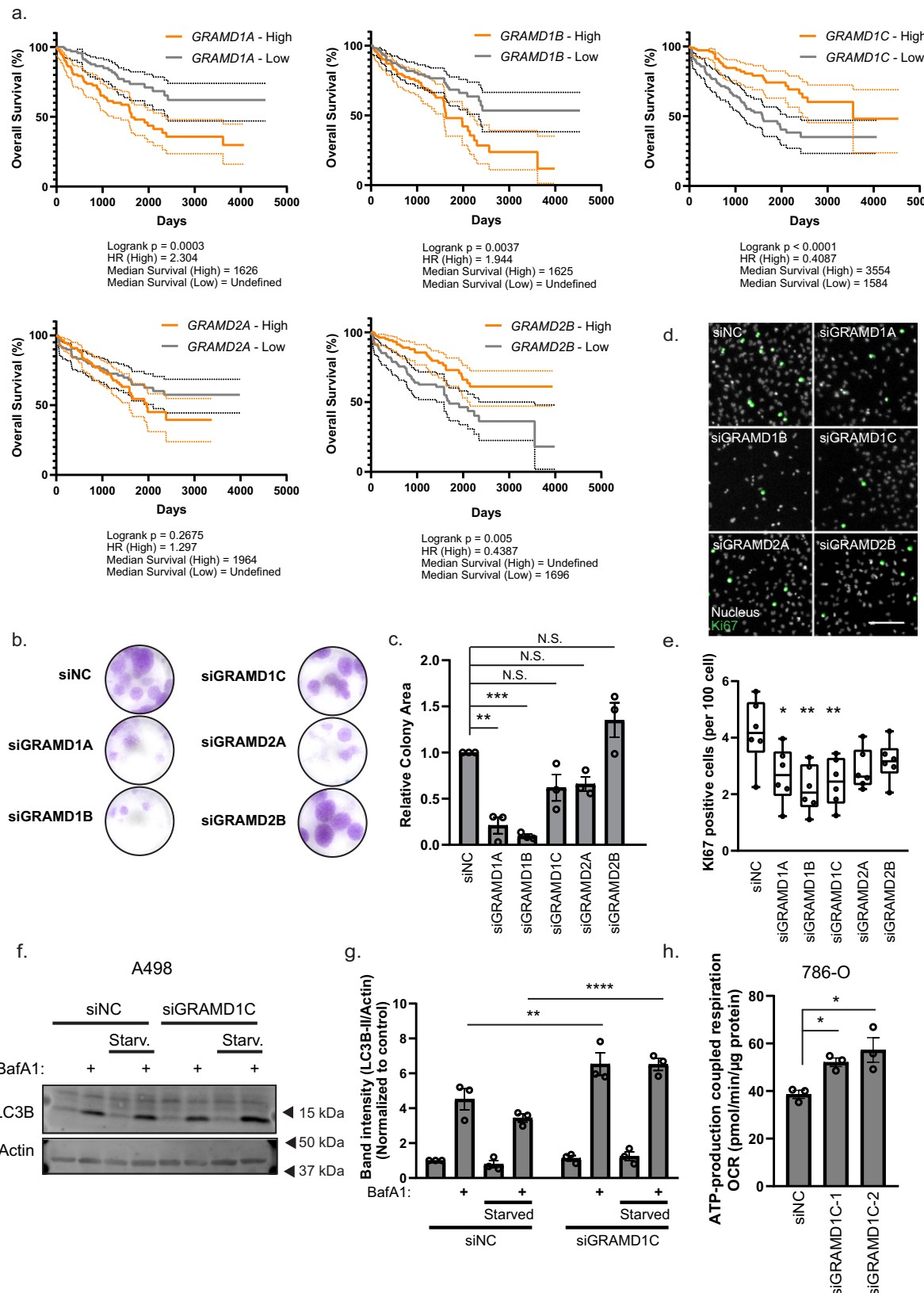

(#A-21206, Invitrogen, 1:500) and Anti-mouse CY3 (#115-165-146, Jackson, 1: 500).

## Materials

The following chemicals were used: Bafilomycin A1 (#BML-CM110, Enzo Life Sciences), CCCP (#BML-CM124, Enzo Life Sciences) Oligomycin A (#S1478, Selleckchem). Antimycin A (#A8674, Sigma Aldrich), DFP (#37940, Sigma Aldrich), DTT (#441496p, VWR), Rotenone (#R8875, Sigma Aldrich), MBCD (#M7439, Sigma Aldrich). For lysis buffers, Complete EDTA-free protease inhibitor (#05056489001, Roche) and PhosStop phosphatase inhibitor (#04906837001, Roche) were used. For live cell imaging, Mitotracker Red (#M22425, Thermo

**Fig. 7 | The GRAMs are involved in ccRCC survival. a** Samples from The Cancer Genome Atlas Kidney Renal Clear Cell Carcinoma (TCGA-KIRC) study were stratified based on *GRAM* expression. Overall survival of samples with high *GRAM* expression (upper quartile, orange line) were compared to low *GRAM* expression (lower quartile, gray line). The dotted lines represent 95% confidence interval. *P* values were obtained using Log-rank (Mantel−Cox) test. **b** For the colony-formation assay, 786-O cells were treated with the indicated siRNAs and incubated for 3 weeks prior to fixation and staining with crystal violet stain. **c** Colony area was quantified and normalized to control siRNA (siNC). Significance was determined using one-way ANOVA followed by Dunnett's comparison test from *n* = 3 independent experiments. Error bar = SEM. *P* value = 0.001 and 0.0003. **d** 786-O cells were treated with siRNA against the indicated GRAMs for 72 h prior to fixation and immunostaining for Ki67. **e** Ki67 positive cells per 100 cells were quantified from 6 independent experiments and represented in a min-max box and whisker plot. The

error bars represent max and min points, while the horizontal lines in the box plot represent the upper quartile, median and lower quartile. Significance was determined using one-way ANOVA followed by Dunnett's multiple comparisons test. Scale bar = 200 μm. *P* value = 0.0237, 0.0021 and 0.0073. **f** A498 cells were treated with siNC or siGRAMD1C for 72 h. The cells were then incubated in DMEM or EBSS ± 100 nM BafA1 for 2 h prior to protein isolation and western blot analysis using the indicated antibodies. **g** Significance was determined using two-way ANOVA followed by Sidak's multiple comparisons test from *n* = 3 experiments. *P* value = 0.0056 and <0.0001. **h** ATP-linked respiration, calculated from the difference between the maximal respiratory capacity and the proton leak from Seahorse analysis, in 786-O cells treated with control (siNC) or siRNA against GRAMD1C for 72 h. Significance was determined using one-way ANOVA followed by Dunnett's comparison test from *n* = 3 experiments. Error bar = SEM. *P* value = 0.049 and 0.0129. Source data are provided as a Source Data file.

Fisher) was used. For the measurement of ROS, CellRox (#C10422, Thermo Fisher) was used. To measure mitochondrial membrane potential, cells were incubated in TMRE (#T669, Thermo Fisher). For amino acid starvation, cells were cultured in Earle's Balanced Salt Solution (EBSS) (Invitrogen). Percoll (#sc-500790A, Santa Cruz). For high-throughput wide-field microscopy, cells were cultured in μ-Plate 96 Well ibiTreat (#89626, Ibidi).

### Cell lines
U2OS (HTB-96) and HEK293T cells (CRL-3216) were cultured in complete DMEM containing 10% v/v fetal bovine serum and 100 U/mL penicillin + 100 μg/mL streptomycin at 37 °C in 5% $CO_2$. 786-O cells (CRL-1932) and A498 (HTB-44) were grown in RPMI and EMEM medium, respectively, supplemented with 10% v/v fetal bovine serum and 100 U/mL penicillin + 100 μg/mL streptomycin at 37 °C in 5% $CO_2$. U2OS TRex FlpIn cells (kindly provided by Steve Blacklow, Harvard Medical School, US) were used for the generation of stable inducible cell lines.

### Lentivirus production and stable cell line generation
Stable cell lines were generated using lentiviral transduction and the FlpIn Trex system[76]. Target genes were cloned into pLenti-III or pLVX viral expression vectors, which were co-transfected with psPAX2 and pCMV-VSVG into HEK293FT cells to generate lentiviral particles. The lentiviral particles were then concentrated using Lenti-X. The resulting lentivirus solution was added to cells and supplemented with 8 μg/mL polybrene. The cells were then selected using the appropriate antibiotics (Puromycin (#p7255, Sigma Aldrich) or Zeocin (#R25001, Invitrogen)). Cells stably expressing inducible mCherry-EGFP-LC3b and MLS-mCherry-EGFP were supplemented with media containing 100 ng/ml Doxycycline. At 72 h, the cells are then treated as described in the figure legends.

An overview of the primers used for cloning and the plasmids used in this study has been included in Supplementary Tables 1 and 2, respectively.

### Knockout cell line generation and validation
GRAMD1C knockout cells were generated using the PX459 system[77]. In short, U2OS cells were transfected with the PX459 vector expressing guides against GRAMD1C. Guide sequences were designed using CHOPCHOP[78] (sequences included in Supplementary Table 1). 24 h post transfection, transfected cells were selected with 3 μg/ml puromycin for 72 h. Single-cell clones are then selected using limited dilution into 96-well tissue culture plates. Due to a lack of an antibody that recognizes endogenous GRAMD1C, validation of the knockout clones was done by sequencing of the relevant region of GRAMD1C from genomic DNA. A total of 15 sequencing reactions were done, all of which indicated a frameshift mutation (E245S*fs20) (Supplementary Fig. 1g).

### siRNA knockdown
siRNA-mediated knockdown was performed using reverse transfection of siRNA against the target gene at a final concentration of 10 nM per oligonucleotide. siRNAs were delivered using Lipofectamine RNAi max (Invitrogen). After 24 h, the cells were washed and replenished with normal media. Silencer Select siRNA (Thermo Fisher) were used against the following target genes. GRAMD1A (s33529), GRAMD1B (s33113), GRAMD1C (s29400 siGRAMD1C-1, s29401 siGRAMD1C-2), GRAMD2a (s47069), GRAMD2b (s35302), OPA1 (s9851), DRP1 (s19559) and Negative Control (s813). Due to the lack of reliable antibodies for endogenous GRAMD1C, knockdown was validated using qPCR against GRAMD1C (Supplementary Fig. 1e, f).

### cDNA synthesis and RT-PCR
RNA was isolated using the RNeasy plus kit (Qiagen) according to the manufacturer's instructions. RNA integrity was confirmed by agarose gel prior to cDNA synthesis. cDNA was synthesized using SuperScript II reverse transcriptase (Thermo Fisher) and real-time quantitative PCR (qPCR) was carried out using SYBR Green Real Time PCR master mix (Qiagen). Normalization of target genes was done against TATA-box-binding protein (TBP) using the $2^{-\Delta\Delta Ct}$ method. The sequences of the qPCR primers used in this study have been included in Supplementary Table 1.

### Flow cytometry
U2OS cells were transfected with siRNA as described above. Upon 48 h post transfection, cells were plated in 12-well plates and left in the incubator O/N. Upon 72 h post transfection, cells were treated with CellRox (#C10422, Thermo Fischer) for 10 or with Tetramethylrhodamine (TMRE) (#T669, Thermo Fisher) for 30 min, according to the manufacturer's instructions. After washing cells were trypsinized, washed in PBS twice, and analyzed using the BD™ LSR II flow cytometer. A total of three experiments in duplicates were performed, and the fluorescent signal was analyzed using FlowJo.

### Wound-healing assay
768-O renal carcinoma cells were transiently transfected as described above. Transfected cells were seeded at 96-well plates (#4379, Essen Bioscience) 48 h post transfection. A total of $4 \times 10^4$ cells/well were seeded in triplicates at approximately 100% well density. At 72 h post transfection one scratch per well was made using the Incucyte® 96-well WoundMaker Tool (#4563, Sartorius). The plate was then loaded in the Incucyte incubator. One image every 20 min for a total of 24 h was acquired for each well. Results were analyzed using the Integrated Cell Migration analysis module (#9600-0012, Sartorius).

### Microscopy and sample preparation
Cells were seeded on glass coverslips or onto glass bottom 96-well imaging plates and treated as indicated. The cells were then washed

twice in prewarmed PBS prior to the addition of warmed fixation solution (3.7% PFA, 200 mM HEPES pH 7.1) and incubated at 37 °C for 20 min. The fixed cells were then washed three times with PBS. Cells destined for immunofluorescence staining were permeabilized with 0.2% NP-40 in PBS for 5 min. The cells were then washed twice in PBS and incubated in 5% BSA in PBS for 30 min. The cells were incubated at 20 °C for 1 h in primary antibody diluted in 5% BSA in PBS. The cells were then washed three times with PBS and incubated in a secondary antibody diluted in 5% BSA in PBS at 20 °C for 45 min. The samples were then washed with PBS. Coverslips were mounted on cover slides using Prolong Diamond Antifade Mounting Solution, and wells of the 96-well imaging plate were filled with PBS to prevent cells from drying out.

Quantitative spot counting of ATG13, ATG13, WIPI2, and LC3 immunostained cells was carried out using a Zeiss AxioObserver microscope (Zen Blue 2.3 Zeiss) fitted with a ×20 Objective (NA 0.5). The samples were illuminated using a solid-state light source (Colibri 7) and multi-bandpass filters (BP425/30, 534/50, 688/145). Imaging of cells expressing MLS-mCherry-EGFP, and mCherry-EGFP-LC3b, was done using the ImageXpress Micro Confocal (Molecular Devices) using a ×20 objective (NA 0.45). Confocal images were taken using the Zeiss LSM 800 microscope (Zen Black 2012 SP5 FP3, Zeiss) equipped with at ×63 oil immersion objective (NA 1.4). Samples were illuminated using a laser diode (405 nm), AR-Laser Multiline (458/488/514 nm), DPSS (561 nm), and HeNe-Laser (633 nm). Live cell confocal imaging was done with cells in a humidified chamber at 37 °C supplemented with 5% $CO_2$ on the Dragonfly (Oxford Instrumentals) with a ×60 objective (NA 1.4) using a EMCCD camera. For live cell imaging, the cells were treated as indicated in the figure legend, before replacing the culture media with FluoroBrite DMEM (#A1896701, Thermo Fisher).

### Bioimage analysis

ATG16, ATG13, WIPI2, and LC3 puncta were quantified using the Cell-Profiler software (2.2.0, 3.1.9, and 4.07, Broad Institute)[79,80]. The nuclei were determined using manual thresholding and object identification of the nuclear stain, and the cells were defined based on a set distance from the center of the nuclei and were confirmed by comparing to the background cytosolic staining of the other channels. Puncta were determined using manual thresholding, object enhancement and object identification. For analysis of mCherry-EGFP-LC3b and mCherry-EGFP-MLS cells, red-only structures were determined by weighting the red signal to match the green signal and by dividing the weighted red signal by the green signal using the CellProfiler software (2.2.0, 3.1.9 and 4.07, Broad Institute). Values that are larger than 1 will represent mitochondria/LC3 structures that have a stronger red signal compared to the green signal. The resulting analysis was manually compared to the image to confirm the accuracy of the imaging pipeline. A value of 1.5 corresponds to twice the signal of red compared to green.

### Western blotting

Cells were treated as indicated in the figure legends before being washed twice in ice-cold PBS. The cells were then lysed in NP-40 lysis buffer (50 mM HEPES pH 7.4, 150 mM NaCl, 1 mM EDTA, 10% glycerol, 0.5% NP-40, Phosphatase inhibitor and Complete Protease inhibitor Cocktail (Roche)). The protein concentration of the lysates was measured with BCA assay (Thermo Fisher). The lysates were run on an SDS-PAGE at 20–30 µg of protein per well before transfer to a PVDF membrane. Blocking was done using a PBS blocking solution (Licor). The resulting membrane was then incubated using the specified primary and secondary antibodies. Visualization of the bound far-red secondary antibodies was performed using the Odyssey CLx imaging system (Licor), and densitometric quantification was performed using the ImageStudio Lite software (Licor).

### Electron microscopy

U2OS cells treated with siNC or siGRAMD1C were grown on poly-l-lysine coated sapphire discs and high-pressure frozen using a Leica HPM100. Freeze substitution was performed as follows: sample carriers containing one sapphire disc each were transferred to individual cryo vials containing 1 ml of freeze substituent (1% OsO4 (w/v), 0.5% uranyl acetate, 0.25% glutaraldehyde in acetone containing a final concentration of 1% $H_2O$) and placed in a temperature-controlled AFS2 (Leica). Freeze substitution occurred at −90 °C for 29 h before the temperature was raised to −60 °C over a time span of 6 h. The samples were kept −60 °C for 2 h before stepwise increasing the temperature to first −20 °C (over 2 h) followed by a temperature raise to +4 °C within 30 min. The samples were kept at 4 °C for 30 min with agitation before transferring the sapphire discs to new cryo vials containing acetone. After three washes with acetone (10 min each), samples were stepwise infiltrated with increasing amounts of epon in acetone (25% (15 min), 50% (30 min), 75% (1 h), 100% (1 h)). The samples were infiltrated in three rounds with fresh, 100% epon over the next 24 h before allowing epon polymerization at 60 °C for 72 h.

Serial sections (250 nm) were cut on an Ultracut UCT ultra-microtome (Leica, Germany) and collected on formvar coated slot grids. Electron micrographs were collected in a Thermo Scientific™ Talos™ F200C microscope equipped with a with a Ceta 16 M camera.

### Oxygen consumption rate measurement

U2OS cells resuspended in complete DMEM were seeded into Seahorse XFe24 Cell Culture microplates at a concentration of $3.5 \times 10^4$ cells per well. The plate was incubated in a humidified incubator at 37 °C for 12 h. The media was then replaced with DMEM without Sodium Bicarbonate (pH 7.4) before analysis with the Seahorse XFe24 Analyzer according to the manufacturer's instructions (XF mito stress test, Agilent). DMEM containing specific mitochondrial inhibitors were loaded into the injector ports of the Seahorse Sensor Plates to obtain the following final concentrations per well (CCCP: 1 µM, Oligomycin: 1.5 µM, Rotenone: 0.5 µM, Antimycin A: 0.5 µM). After the analysis, the cells were washed in ice-cold PBS and lysed for protein quantification using BCA Assay (Thermo Fisher). Quantification was conducted on the Seahorse Analytics software (seahorseanalytics.agilent.com, Agilent), using the measured protein concentration from each well for normalization.

### Mitochondria isolation

Mitochondria were isolated using two different methods. For percoll density gradient isolation, cells are scraped with ice-cold mitochondrial isolation buffer (5 mM Tris-HCl pH 7.4, 210 mM mannitol, 70 mM Sucrose, 1 mM EDTA, 1 mM DTT, 1X PhosStop, 1x PIC) and mechanically lysed using a cell homogenizer (Isobiotech) equipped with a 16-µm clearance ball by passing the cell suspension 10 times through the homogenizer. The resulting solution was then centrifuged at 1500×$g$ for 5 min at 4 °C to pellet nucleus and unbroken cells. The suspension was then centrifuged at 14,000×$g$ for 20 min to obtain a crude mitochondrial pellet. The pellet was then resuspended in mitochondrial isolation buffer and layered above a premade percoll gradient of 50%, 22%, and 15% in a 5 ml ultracentrifuge tube. The tube was then centrifuged at 30,000×$g$ for 1 h. A white layer between the 50% and 22% gradient is isolated using a syringe and needle. Percoll was separated from the isolated mitochondrial fraction by washing in mitochondrial isolation buffer and centrifugation at 14,000×$g$ for 15 min for 4−5 times. After the final wash, the pellet containing isolated mitochondria was lysed with RIPA lysis buffer.

For affinity purification of mitochondria, mitochondria were isolated from cells stably expressing 3xHA-EGFP-OMP25 according to ref. 81 with minor modifications. In short, cells were scraped in ice-cold KPBS (136 mM KCl, 10 mM KH₂PO₄, pH 7.25) and mechanically lysed using a cell homogenizer (Isobiotech) equipped with a 16 µm clearance

ball by passing the cell suspension ten times through the homogenizer. The resulting solution is then centrifuged at 1500×*g* for 5 min at 4 °C to pellet nucleus and unbroken cells. The supernatant was then incubated with anti-HA magnetic beads (Thermo Fisher) for 5 min, before washing with KPBS and resuspension in 2× SDS-Page loading buffer.

## Cholesterol quantification
Cholesterol quantification was done using the Cholesterol/Cholesteryl Ester Assay Kit (Abcam, ab65359). Briefly, mitochondria were isolated using differential centrifugation. The resulting mitochondrial pellet was then resuspended with Chloroform:Isopropanol:NP-40 (7:11:0.1) to extract lipids. The mixture was then air-dried at 50 °C to remove the chloroform. The resulting lipids were analyzed according to the manufacturer's instructions. The resulting values were normalized to proteins measured by BCA assay.

## Mitochondrial cholesterol mCherry-D4 assay
Isolated mitochondria were incubated in a Mitochondria Isolation Buffer 100 μg/mL of mCherry-D4 ± 5 mM MBCD at 37 °C for 30 min. The mitochondria were then washed three times in mitochondria isolation buffer and lysed with 2× SDS-page loading buffer and immediately subjected to western blot analysis.

## Mitochondrial structure classification
Cells stably expressing IMLS were treated with siRNA against the GRAMs, OPA1, and DRP1. After 72 h of knockdown, the cells were fixed and imaged. In these cells, only DAPI and the EGFP signals were measured. The mitochondrial intensity distribution, texture, shape, and area were measured using CellProfiler. The results were used in Cell-Profiler Analyst (v2.2.1, Broad Institute)[82,83] to classify mitochondrial morphology. Mitochondria of siOPA1 and siDRP1 cells represented fragmented and tubular phenotypes, respectively. The classifier was trained with a confusion matrix >0.90 for each phenotype.

## Long-lived protein degradation (LLPD)
Cells were incubated in complete DMEM supplemented with 0.25 μCi/mL L-$^{14}$C-valine (Perkin Elmer) for 24 h. The radioactive media was then removed, and the cells were washed three times with complete DMEM supplemented with 10 mM L-valine, and finally chased for 16 h in complete DMEM supplemented with 10 mM L-valine. The cells were then washed three times in PBS and either starved in EBSS or not for 4 h, in the presence or absence of 100 nM BafA1. The supernatant was collected into tubes containing 15% trichloroacetic acid before subsequent incubation at 4 °C for 12 h. The cells remaining in the dish were lysed with 0.2 M KOH. The supernatant was recovered by centrifugation. The supernatant and the cell lysate were added into separate scintillation tubes containing Ultima Gold LSC cocktail (Perkin Elmer) and the radioactivity was measured by a TriCarb 3100TR liquid scintillation counter (Perkin Elmer). Long-lived protein degradation was calculated by dividing the radioactivity in the supernatant fraction by the total radioactivity in both the supernatant and cell lysate.

## Co-IP mass spec
U2OS cells expressing GRAMD1C-EGFP, GRAMD1C(ΔGRAM)-EGFP or EGFP were lysed with NP-40 lysis buffer (150 mM, 1.0% NP-40, 50 mM Tris-HCl pH 8.0) supplemented with PhosStop phosphatase inhibitor (Sigma) and complete protease inhibitor cocktail (Sigma). Co-immunopurification was then carried out using EGFP-TRAP (Chromotek) according to the manufacturer's instructions. The resulting beads with the coprecipitated proteins were washed twice with 50 mM ammonium bicarbonate. Proteins on beads were reduced and alkylated and further digested by trypsin overnight at 37 °C. Digested peptides were transferred to new tube, acidified, and the peptides were de-salted for MS analysis.

## LC-MS/MS
Peptides samples were dissolved in 10 μl 0.1% formic buffer and 3 μl loaded for MS analysis. LC-MS/MS analysis of the resulting peptides was performed using an Easy nLC1000 liquid chromatography system (Thermo Electron, Bremen, Germany) coupled to a QExactive HF Hybrid Quadrupole-Orbitrap mass spectrometer (Thermo Electron) with a nanoelectrospray ion source (EasySpray, Thermo Electron). The LC separation of peptides was performed using an EasySpray C18 analytical column (2-μm particle size, 100 Å, 75-μm inner diameter and 25 cm; Thermo Fisher Scientific). Peptides were separated over a 90 min gradient from 2% to 30% (v/v) ACN in 0.1% (v/v) FA, after which the column was washed using 90% (v/v) ACN in 0.1% (v/v) FA for 20 min (flow rate 0.3 μL/min). All LC-MS/MS analyses were operated in a data-dependent mode where the most intense peptides were automatically selected for fragmentation by high-energy collision-induced dissociation.

Raw files from LC-MS/MS analyses were submitted to MaxQuant 1.6.17.0 software[84] for peptide/protein identification. Parameters were set as follow: Carbamidomethyl (C) was set as a fixed modification and PTY; protein N-acetylation and methionine oxidation as variable modifications. First search error window of 20 ppm, and the mains search error of 6 ppm. Trypsin without proline restriction enzyme option was used, with two allowed miscleavages. Minimal unique peptides were set to one, and FDR allowed was 0.01 (1%) for peptide and protein identification. The Uniprot human database was used. Generation of reversed sequences was selected to assign FDR rates. Further analysis was performed with Perseus[85], limma[86], Package R[87]. Volcano plots were plotted with EnhancedVolcano[88]. The gene ontology (GO) analysis was performed with shinyGO[89].

## TCGA data
Survival data and cancer stage from all samples in the TCGA-KIRC (clear cell renal cell carcinoma, data release 27.0) cohort was downloaded with the TCGABiolinks R package[90]. Survival curve comparisons were carried out in GraphPad Prism 8.0.1 using Log-rank (Mantel−Cox) test.

## Sequence alignment
The GRAM domain sequences of each GRAM protein were obtained from Uniprot, which were then aligned using Clustal Omega[91] and BlastP[92].

## Crystal violet staining
786-O cells were seeded into 6-well and 24-well plates in quadruplicates and treated as indicated. After 3 weeks, the cells are fixed in a staining solution (6% Glutaraldehyde, 0.5% Crystal Violet) for 1 h at room temperature. The fixation solution was removed, and the cells were rinsed by multiple gentle immersion in H$_2$O. The stained cells were then imaged on a BioRad ChemiDoc MP analyzer. Quantification of the colony area was done using ImageJ software.

## Statistics and significance
Statistical analysis was carried out using Prism (8.01) using the test as indicated in the figure legends. All relevant statistical tests are described in the figure legends, and all data values come from distinct samples. ****$P < 0.0001$, ***$P < 0.001$, **$P < 0.01$, *$P < 0.05$ or N.S. = not significant.

## Reporting summary
Further information on research design is available in the Nature Research Reporting Summary linked to this article.

## Data availability
All data are available upon reasonable request. The proteomics data generated in this study have been deposited in the PRIDE database

under accession codes PXD033125 (Supplementary Data 1) and PXD027502 (Supplementary Data 2). Source data are provided with this paper.

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

## Acknowledgements

This work was supported by the Research Council of Norway through its Centre of Excellence funding scheme (Project: 262652, A.S.) and FRIPRO grant (Project: 249753, M.N., C.C. and A.S.), the Norwegian Cancer Society (Project: 171318, M.J.M and A.S.), the European Union's Horizon 2020 research and innovation programme under the Marie Skłodowska-Curie grant agreement No. 801133 and the UiO Scientia Fellow program —co-funding of Regional, National and International Programmes (COFUND, A.S. and S.Singh) and the MSC ETN grant Agreement No. 765912 (DRIVE, A.S., and A.L.). The authors would like to thank Sonia Peña Pérez for help with the graphical abstract, Laura Rodriguez de la Ballina for technical help and constructive feedback, and Helene Knævelsrud for the A498 cells. The authors would also like to thank Sachin Singh, Maria Stensland, and Tuula Anneli Nyman for their assistance with the mass-spectrometry-based proteomic experiments. Mass-spectrometry-based proteomic analyses were performed by the Proteomics Core Facility, Department of Immunology, University of Oslo/Oslo University Hospital, which is supported by the Core Facilities program of the South-Eastern Norway Regional Health Authority. This core facility is also a member of the National Network of Advanced Proteomics Infrastructure (NAPI), which is funded by the Research Council of Norway INFRASTRUKTUR-program (Project number: 295910). We would like to thank The Norwegian Core Facility for Human Pluripotent Stem Cells at the Norwegian Center for Stem Cell Research for mycoplasma testing of our cells, and the Advanced Light Microscope Facility at the University of Oslo for access to their microscopes.

## Author contributions

M.N., C.C., and A.S. conceived and planned the experiments. S. Singh carried out the analysis of mass spec data. S.Schultz carried out the electron microscopy experiments. M.N., C.C., A.L., and L.T.M. performed the experiments and analyzed data. M.N. and A.S. wrote the manuscript with input from all authors. S.N. provided TCGA data and contributed to the interpretation of the results. M.J.M. provided critical feedback on the manuscript. All authors discussed the results and contributed to the final manuscript.

## Competing interests

The authors declare the following competing interests: M.J.M is an employee of AstraZeneca. All other authors declare no competing interests.
