## [Peer Review File · Nature Communications]

The cholesterol transport protein GRAMD1C regulates autophagy initiation and mitochondrial bioenergeticsReviewer #1 (Remarks to the Author):

Summary:

In this manuscript, Ng et al. explored the role of cholesterol in starvation induced autophagy. They reported that fast depletion of cholesterol using methyl- β cyclodextrin (MBCD) followed by amino acid starvation increases LC3B lipidation and puncta formation, which was also true for long term cholesterol depletion using atorvastatin (ATV). Further they showed that autophagosome biogenesis is also increased upon cholesterol depletion. In-order to understand the mechanism of autophagy biogenesis in cholesterol depletion and starvation, they explored the role of mTORC1 substrate p70S6K and ULK1 component Atg13 in time course studies. Atg13 positive structures were already found before complete inactivation of mTORC1, suggesting the pathway to be mTORC1 independent. To see the effect of cholesterol removal on generation of membrane curvature, they used a reporter based on BATS domain of ATG14L1. The domain is known to bind to PI(3)P and colocalize with autophagy initiation proteins. Upon MDCB treatment, EGFP-BATS puncta are increased, thus suggesting that cholesterol depletion induces formation of curved early autophagosome membranes. In order to understand the role of cholesterol transport proteins, they further performed siRNA depletion of GRAM family of proteins and found that only loss of GRAMD1C increases LC3B turnover in starvation. They also explored the role of GRAMD1C in mitophagy. Loss of GRAMD1C yielded increased membrane recruitment of early autophagy components, thereby suggesting GRAMD1C to be a negative regulator of autophagy initiation. They further investigated the importance of GRAM domain in GRAMD1C protein in establishing a mitochondrial contact site using membrane fractionation and proteomics studies. Since GRAMD1C was found to associate with many mitochondrial proteins via its GRAM domain, mitochondrial bioenergetics and cholesterol transport to mitochondria were studied in GRAMD1C depletion. They conclude that GRAMD1C acts as a negative regulator of mitochondrial cholesterol abundance and facilitate cholesterol transport from mitochondria to ER. Finally, they explored the role of GRAM proteins in clear cell renal carcinoma(ccRCC) patients using gene expression data from TCGA KIRC cohort and in ccRCC patient derived cells. They conclude that GRAM family proteins contribute to overall survival of ccRCC patients possibly by regulating cancer cell survival. Overall, this article provides an interesting connection between cholesterol transport and autophagy initiation with a further scope to explore importance of autophagy initiation as a key factor in ccRCC patient survival.

Major points:

- 1) Figure 3 a: The authors should provide a better representation of siGRAMD1C which clearly shows increased red puncta compared to control in starved condition.
- 2) Figure 4 h: The authors should provide a better example of EGFP BATS puncta in si RNA treatments showing clear difference from control.
- 3) Figure 3 and 4: The authors need to show a Western blot and/or qPCR to show the knockdowns of GRAM proteins or knockout of GRAMD1C. For knockout studies, the authors should validate their key findings with a second cell clone.
- 4) Supplementary figure 2: The authors need to provide an additional lysosomal staining to really confirm the localization of GRAMD1C in starved conditions.
- 5) Supplementary Figure 3 b and d: The style of these two graphs looks quite different than graphs from other figures. Hence, it would be easier for the reader if they are all in similar format.
- 6) No evidence was provided to show indeed GRAMD1C-EGFP localized to ER as stated in line 242.
- 7) Figure 5 d: The authors should analyze in which cell fraction the wild-type GRAMD1C and delta GRAM co-fractionate..

8) Figure 6 a: It is not clear what only D4 lane is. Is it just to show the expression of D4? If so, what cell fraction is it in? What are the double bands?

9) Figure 6d: Why is there such a variation in the OCR between two si RNAs of GRAMD1C?

10) Figure 6d, Sup 4f, g: The authors should explain acronyms such as OCR, TMRE, CellRox in the results text.

11) The authors should examine the effects of (i) EGFP-GRAMD1C Δ GRAM and EGFP-GRAM domain in some of the autophagy or mitochondria bioenergetics assays and (ii) whether the EGFP GRAM domain only can rescue some of the effects in GRAMD1C knockouts.

Minor points:

12) Minor typos: Line 103: "leads", line 419 and 420: "found" instead of "find"

13) Line 277: explain what SREBP is.

Reviewer #2 (Remarks to the Author):

Summary:

Ng et al. revisited the effect of cholesterol depletion on autophagy and showed that cholesterol depletion by either MBCD or statin promoted autophagy initiation. They then investigated the potential role of an evolutionarily conserved family of ER-anchored lipid transfer proteins, GRAMD1s (GRAMD1A/1B/1C), in the regulation of autophagy and found that the depletion of one of the GRAMD1s (GRAMD1C) facilitated starvation-induced autophagy (possibly by modulating autophagy initiation). Using live cell imaging and biochemical assays, they found that a small fraction of GRAMD1C localized to ER-mitochondria contact sites and that GRAMD1C interacted with mitochondria possibly through its GRAM domain. They were not able to show which mitochondrial protein was responsible for the targeting of GRAMD1C to mitochondria. To further investigate the function of GRAMD1C at ER-mitochondria contact sites, the authors depleted GRAMD1C expression in U2OS cells by RNAi or genetic knockout and found that this manipulation lead to accumulation of cholesterol in mitochondria and enhanced mitochondrial oxidative phosphorylation. Based on these results, they concluded that GRAMD1C functions at ER-mitochondria contacts to regulate mitochondrial bioenergetics. They then investigated the relationship between the expression of GRAMD1s (as well as other proteins that contain the GRAM domain without the sterol-harboring StART-like domain, GRAMD2A, GRAMD2B) and the survival of patients suffering from clear cell renal cell carcinoma and reported that there is some correlation between GRAMD1 expression and patient survival.

Overall, this study provides little mechanistic insights into the role of cholesterol in autophagy. It also remains unclear how GRAMD1C affects mitochondrial bioenergetics. Most importantly, the three major pieces of data that the authors presented in this study, namely 1) effects of cholesterol depletion in autophagy, 2) the potential role of GRAMD1C in the regulation of mitochondrial cholesterol levels and mitochondrial oxidative phosphorylation at ER-mitochondria contact sites, and 3) the association between the severity of clear cell renal cell carcinoma and the expression levels of various GRAMD family proteins, are not well connected to support each other. The authors should narrow their focus of the study and perform more in-depth analysis of selected data sets. With the current form, the study remains descriptive and preliminary for publication.

Major comments:

1. How MBCD affects autophagy remains unclear. It has been known that depletion of cholesterol by 1-hour treatment of MBCD induces autophagy (e.g., PMID: 17056010). If it is independent of mTORC1 as the authors claimed in their current study, what is the molecular mechanism that underlies autophagy induction caused by cholesterol depletion? Authors showed increased appearance of EGFP-BATS puncta both in starved and MBCD-treated conditions in Figure 2h; these results suggest that starvation and cholesterol depletion cause autophagy induction with similar mechanisms rather than distinct mechanisms (opposite to what the authors proposed). Another important issue is non-selective action of MBCD for cholesterol depletion; MBCD affects the levels of cholesterol in the PM (where the majority of cellular cholesterol is present) as well as lysosomes, and it might have indirectly affected autophagy induction.

2. This study provides limited insights into how GRAMD1C affects autophagy. Previous studies had shown that another member of GRAMD1s, GRAMD1A, regulates autophagy (PMID: 31222192). In this study, the authors provided some evidence that supports a role for GRAMD1C as a negative regulator of autophagy (Figures 3 and 4). They also reported that GRAMD1C potentially localized to ER-mitochondria contacts and regulated mitochondrial cholesterol levels as well as mitochondrial oxidative phosphorylation (Figure 5). However, how the function of GRAMD1C at ER-mitochondria contacts is related to the regulation of autophagy induction was not addressed.

3. Related to the above-mentioned point, the model/conclusion for the function of GRAMD1C proposed by the authors is not well supported. The authors stated in the line 231 "these results indicate that GRAMD1C regulates autophagosome biogenesis by removal of cholesterol from ER membranes". However, there is no data showing that GRAMD1C removes cholesterol from ER membranes in this study.

4. The authors suggest that GRAMD1C transports cholesterol from mitochondria to the ER at ER-mitochondria contacts, but the evidence supporting this function is weak. More experiments are needed to support a direct role of GRAMD1C in cholesterol transport at ER-mitochondria contacts. Related to this point, how 1) enhanced mitochondrial oxidative phosphorylation (Figure 6d) is caused by 2) elevated levels of cholesterol in mitochondria (Figure 6a-c) in cells lacking GRAMD1C expression is not addressed. It is also unclear how these alterations in mitochondria affect autophagy induction.

5. Finally, the reported association between the expression of GRAMDs and clear renal cell carcinoma is interesting but preliminary and largely descriptive, providing limited mechanistic insights and no link to autophagy induction.

Other important issues:

1. No rescue experiments were performed throughout this study, making it hard to determine whether the phenotypes reported in this study were truly due to the loss/reduction of a cholesterol transporting function of GRAMD1C.

2. The localization analysis of GRAMD1C (Figure 5) is premature and their statement in line 262 "Thus, our results show that GRAMD1C interacts with the mitochondria through its GRAM domain" is not well supported. They should at least show GRAMD1C lacking the GRAM domain does not interact with mitochondria.

3. It is also unclear which one of the mitochondrial proteins is directly involved in recruiting GRAMD1C to mitochondria (Figure 6e, 6g). The authors should verify the mass spec results with Western Blotting for at least some of the proteins that they identified through IP-MS analysis. They did not find GRAMD1C in anti-EGFP immunoprecipitates from cells expressing GRAMD1C lacking the GRAM domain (i.e., GRAMD1C Δ GRAM) (Table 1b), which needs to be explained or revisited.

4. The use of D4 to examine the levels of mitochondrial cholesterol is not well validated

in this study. There is very little reduction of D4 binding to mitochondria upon MBCD treatment (Figure 6a).

5. It is unclear why the abundance of cholesterol-associated proteins (STARD9, ERLIN etc.) is altered in GRAMD1C depleted cells (line 278-279).

Reviewer #3 (Remarks to the Author):

In their manuscript, Matthew Yoke Wui Ng and colleagues report that the ER-located protein GRAMD1C specialized in cholesterol transport is a negative regulator of autophagosome biogenesis. This elegant study is based on robust cell biology and biochemistry readouts and deciphers the molecular insights of GRAMD1C function at ER/mitochondria membranes interface. This work shed light on the functional importance of low-cholesterol membrane compartments in autophagy early step regulation. The goal of the study is clearly stated, the manuscript is well written and data are well presented and explained. The physiological relevance brought by the implication of GRAM family proteins in clear cell renal carcinoma (CCR) is of interest, although the direct link of the cellular hallmarks of CCR with autophagosome biogenesis is not very explored. With a dedicated revision, this paper could be of interest for readers of Nature Communications.

I have several remarks concerning the experimental data as well as the general scientific context:

- Concerning the mechanistic explanation of low-cholesterol membrane requirement in autophagosome biogenesis: i) have authors checked for ATG14L and VPS34 behavior in low cholesterol cells and/or in GRAMD1C depleted cells? ii) What happens to ER-PI3P positive membranes associated in low cholesterol cells and/or in GRAMD1C depleted cells?
- Concerning the involvement of GRAM family proteins in CCR: in the light of authors' data about GRAMD1C implication in autophagy regulation, it is worthy to notice that autophagic machinery has been linked to proximal tubule renal epithelial cells differentiation and (mitochondria and lipid) metabolism via primary-cilium and mechanical/shear stress. The authors mentioned that starvation-induced autophagy is negatively regulated by GRAMD1C, but what about shear-stress-induced autophagy?
- The manuscript could benefit from electron microscopy analyzes of "autophagic structures" generated in low cholesterol cells and in GRAMD1C depleted cells, notably concerning the membrane curvature part. In the same line, it could be interesting to decipher whether the boosted set of autophagosomes formed in absence of GRAMD1C are fully functional.
- In the mitochondria fractionation experiments, data could benefit from using empty GFP as a negative control.
- The implication of GRAM/ER-to-plasma membrane cholesterol transfer putative relationship with ER/plasma membrane contact sites (which are also implicated in autophagosome biogenesis) could be discussed.
- It is not clear to me whether "ER-mitochondria contact sites based cholesterol transport regulation" enables direct and local ER membrane lowering of cholesterol, for example at omegasome (other membranes normally positive for cholesterol (such as Golgi or endosomes) have been reported to participate to autophagosome biogenesis as well).

- **Could authors discuss a bit more about the absence of P62 readouts associated phenotypes in their study?**
- **Could authors specify the use of U2OS cells in the context of the study?**
- **I was wondering why western blots of LC3 only displayed LC3-II form?**
- **Reference number 3 is not referring to fatty acids enrichment in autophagosomes membranes.**

Rebuttal NCOMMS-21-25735

“GRAMD1C regulates autophagy initiation and mitochondrial bioenergetics through ER-mitochondria cholesterol transport”

Dear editor,

We would like to thank you and the reviewers for your thorough reading of the manuscript and for the constructive comments and criticism, which have significantly strengthened the manuscript.

We have addressed all the recommendations raised by the reviewers in the point-by-point response below.

Reviewer 1:

Summary:

In this manuscript, Ng et al. explored the role of cholesterol in starvation induced autophagy. They reported that fast depletion of cholesterol using methyl- β cyclodextrin (MBCD) followed by amino acid starvation increases LC3B lipidation and puncta formation, which was also true for long term cholesterol depletion using atorvastatin (ATV). Further they showed that autophagosome biogenesis is also increased upon cholesterol depletion. In-order to understand the mechanism of autophagy biogenesis in cholesterol depletion and starvation, they explored the role of mTORC1 substrate p70S6K and ULK1 component Atg13 in time course studies. Atg13 positive structures were already found before complete inactivation of mTORC1, suggesting the pathway to be mTORC1 independent. To see the effect of cholesterol removal on generation of membrane curvature, they used a reporter based on BATS domain of ATG14L1. The domain is known to bind to PI(3)P and colocalize with autophagy initiation proteins. Upon MDCB treatment, EGFP-BATS puncta are increased, thus suggesting that cholesterol depletion induces formation of curved early autophagosome membranes. In order to understand the role of cholesterol transport proteins, they further performed siRNA depletion of GRAM family of proteins and found that only loss of GRAMD1C increases LC3B turnover in starvation. They also explored the role of GRAMD1C in mitophagy. Loss of GRAMD1C yielded increased membrane recruitment of early autophagy components, thereby suggesting GRAMD1C to be a negative regulator of autophagy initiation. They further investigated the importance of GRAM domain in GRAMD1C protein in establishing a mitochondrial contact site using membrane fractionation and proteomics studies. Since GRAMD1C was found to associate with many mitochondrial proteins via its GRAM domain, mitochondrial bioenergetics and cholesterol transport to mitochondria were studied in GRAMD1C depletion. They conclude that GRAMD1C acts as a negative regulator of mitochondrial cholesterol abundance and facilitate cholesterol transport from mitochondria to ER. Finally, they explored the role of GRAM proteins in clear cell renal carcinoma(ccRCC) patients using gene expression data from TCGA KIRC cohort and in ccRCC patient derived cells. They conclude that GRAM family proteins contribute to overall survival of ccRCC patients possibly by regulating cancer cell survival.

Overall, this article provides an interesting connection between cholesterol transport and autophagy initiation with a further scope to explore importance of autophagy initiation as a key factor in ccRCC patient survival.

We thank the reviewer for these encouraging comments.

Major points:

Figure 3a: The authors should provide a better representation of siGRAMD1C which clearly shows increased red puncta compared to control in starved condition.

We have now replaced Figure 3a with images that better represent the levels of red puncta seen upon quantification of >500 cells per condition from n = 3 experiments, as shown in Figure 3b. This includes another field of view for siGRAMD1C where the LC3 puncta are better defined.

Figure 4h: The authors should provide a better example of EGFP BATS puncta in siRNA treatments showing clear difference from control.

The image representing siNC treated cells in Figure 4h has been replaced with a more representative image to better represent the increase in EGFP-BATS puncta accumulation in siGRAMD1C treated cells, as quantified in Figure 4i from >500 cells per condition from n = 3 experiments.

Figure 3 and 4: The authors need to show a Western blot and/or qPCR to show the knockdowns of GRAM proteins or knockout of GRAMD1C. For knockout studies, the authors should validate their key findings with a second cell clone.

We have tested several antibodies against GRAMD1C (Sigma, Abcam, Biobryt (not shown)). However, these were not capable of identifying endogenous GRAMD1C, but did recognize overexpressed GRAMD1C (and GRAMD1C(Δ GRAM)) in both Wt and GRAMD1C KO (GKO) cells stably expressing untagged GRAMD1C constructs (Figure included for reviewer below). Thus, in order to validate knockdown of GRAMD1C, we routinely observed mRNA degradation by qPCR (Supplementary figure 1f). All of the siRNAs used against GRAM family members successfully downregulated target gene

mRNA levels (Supplementary figure 1e). However, we observed poor knockdown efficiency for GRAMD1B and GRAMD2A, likely due to their low expression in U2OS cells or low primer efficiency as the Ct values of GRAMD1B and GRAMD2A were high (>33).

The figure shows antibody testing for lysates from Wt and GKO U2OS cells expressing untagged GRAMD1C or GRAMD1C (Δ GRAM). The predicted band size for GRAMD1C is 75 kDa and 55 kDa for GRAMD1C (Δ GRAM). Endogenous GRAMD1C is not detected by either antibody.

We were unfortunately unable to generate a second knockout clone for GRAMD1C, however we obtained heterozygous GRAMD1C knockout cell lines, referred to as HKO, which were validated by sequencing of the target region on GRAMD1C (Figure included for reviewer below). In the second HKO clone, termed HKO2, 5 out of 7 sequencing runs revealed a frameshift mutation. By western blot, the band at the predicted size (70-75 kDa) potentially corresponding to GRAMD1C was weaker, but as the actin loading control also was weaker we do not believe that this band actually represents GRAMD1C. Unlike GKO cells, HKO2 cells do not significantly alter LC3 lipidation upon starvation compared to their Wt counterpart (b-c). HKO2 cells also have quite similar mitochondrial respiration profile as Wt cells.

The ATP-production linked respiration in HKO was slightly reduced and this was further reduced with the overexpression of ΔGRAM (d-e). We have chosen not to include these data in the manuscript.

a. Western blot analysis of HKO cell lines. The bands at the predicted size of GRAMD1C of around 70 kDa is shown.
b. HKO2 cells stably expressing GRAMD1C, VASt domain mutant GRAMD1C(Ω1), and GRAMD1C(ΔGRAM) were starved for 2 hrs ± 100 nM BafA1 and subjected to western blot analysis for the indicated proteins.
c. Quantification of b.
d. Oxygen consumption rate of HKO2 cells expressing GRAMD1C or GRAMD1C(ΔGRAM).
e. The ATP-production linked respiration of HKO2 cells expressing GRAMD1C or GRAMD1C(ΔGRAM).

Supplementary figure 2: The authors need to provide an additional lysosomal staining to really confirm the localization of GRAMD1C in starved conditions.

We have now costained cells expressing mCherry-EGFP-GRAMD1C with LAMP1 (Supplementary figure 3c), clearly showing red-only GRAMD1C puncta within LAMP1 structures, indicating that GRAMD1C is delivered to the lysosomal lumen. Additionally, overexpressed GRAMD1C-EGFP accumulates when cells are treated with the vacuolar H⁺-ATPase inhibitor BafA1, indicating that GRAMD1C is turned over in the lysosome (Supplementary figure 3d).

Supplementary Figure 3 b and d: The style of these two graphs looks quite different than graphs from other figures. Hence, it would be easier for the reader if they are all in similar format.

The format of these graphs (now Supplementary Figure 2b and d), and all other graphs, have now been unified to make it easier to read.

No evidence was provided to show indeed GRAMD1C-EGFP localized to ER as stated in line 242.

We have now included data showing that GRAMD1C-EGFP localizes to the same structures as the ER protein SEC61-mCherry (Supplementary figure 4b), in line with a previous publication showing that proteins of the GRAM family localize to the ER (PMID: 30220461). Additionally, from our MS data of the GRAMD1C-EGFP interactome, we find GRAMD1C to interact with several ER resident proteins such as Calnexin and TMX1 (Table 1).

Figure 5 d: The authors should analyze in which cell fraction the wild-type GRAMD1C and delta GRAM co-fractionate.

To address this, lysates from GRAMD1C-EGFP and GRAMD1C(Δ GRAM)-EGFP expressing cells were fractionated into cytosol and crude mitochondria fractions. Both GRAMD1C and GRAMD1C(Δ GRAM) were found in the crude mitochondria fraction (Supplementary figure 4g). This is likely explained by the ability of Δ GRAM to interact with other GRAM proteins and mitochondrial proteins (Figure 5f, Table 1). Indeed, we show that the GRAMD1C GRAM domain alone is unable to interact with GRAMD1A and GRAMD1B (Figure 5f). Alternatively, as GRAMD1C and GRAMD1C(Δ GRAM) are ER-anchored proteins, it is difficult to rule out that their presence in the crude mitochondria fraction can be due to contamination from the ER, which is a prominent problem during mitochondria isolation. Indeed, we found traces of ER protein CLIMP63 in the crude mitochondria fraction.

Figure 6 a: It is not clear what only D4 lane is. Is it just to show the expression of D4? If so, what cell fraction is it in? What are the double bands?

We are sorry about the confusion here. In this assay, recombinant mCherry-D4 was added directly to purified mitochondria. To show the size of the mCherry-D4 band, we added purified mCherry-D4 into one of the lanes of the western blot to represent purified mCherry-D4 protein. The size of the lower mCherry-D4 band corresponds to the size of free mCherry, possibly formed from the degradation of

mCherry-D4. Indeed, we do not see free mCherry binding to mitochondria. We have now included an asterisk to indicate free mCherry and labelled the D4 lane with “Rec.D4” to indicate that we added recombinant D4 to the lane to avoid confusion.

Figure 6d: Why is there such a variation in the OCR between two siRNAs of GRAMD1C?

The OCR profiles of the two siRNAs of GRAMD1C are quite similar and both significantly different than the non-targeting control (siNC). Again, we apologize for the confusing labelling of the figure. The symbols alone were difficult to read, and we have now color coded the lines to make it easier for readers to interpret the data (now Figure 6e-f).

Figure 6d, Sup 4f, g: The authors should explain acronyms such as OCR, TMRE, CellRox in the results text.

All acronyms have now been explained. OCR (Figure legend 6e), TMRE (Supplementary figure legend 5d), CellRox (Supplementary figure legend 5e).

The authors should examine the effects of (i) EGFP-GRAMD1C Δ GRAM and EGFP- GRAM domain in some of the autophagy or mitochondria bioenergetics assays and (ii) whether the EGFP GRAM domain only can rescue some of the effects in GRAMD1C knockouts.

We have now evaluated autophagy flux, quantified as LC3B-II relative to actin levels in cells with stable overexpression of GRAMD1C-EGFP, GRAMD1C(Δ GRAM)-EGFP or EGFP-GRAM incubated in control or starvation media in the absence or presence of BafA1 (Supplementary figure 3d-e). Neither overexpression of GRAMD1C, GRAMD1C(Δ GRAM) or EGFP-GRAM significantly alter starvation-induced autophagy flux as compared to control cells expressing EGFP.

To address whether the EGFP-GRAM domain only can rescue some of the effects seen in GRAMD1C knockouts, we first generated stable GKO rescue cells expressing GRAMD1C-EGFP, GRAMD1C(Δ GRAM)-EGFP or EGFP-GRAM. Neither of the rescue constructs were able to rescue the autophagy or mitochondrial phenotypes observed in GKO (or GRAMD1C depleted) cells, indicating that the EGFP tag might prevent the function of GRAMD1C. We therefore generated untagged GRAMD1C lenti constructs that were transduced in GKO cells to generate stable rescue cell lines. Indeed, the full length GRAMD1C was now able to rescue the autophagy and respiration phenotypes observed in GKO cells (Figure 3f-g and Figure 6g-h). Importantly, the GRAM domain was found to be important for rescue of the autophagy phenotype, as GKO cells expressing GRAMD1C(Δ GRAM) demonstrated increased LC3B lipidation (as seen in GKO cells) (Figure 3f-g). In contrast, both GRAMD1C and GRAMD1C(Δ GRAM) were able to rescue the increased ATP-linked respiration seen in GKO cells (Figure 6g-h).

Minor typos: Line 103: “leads”, line 419 and 420: “found” instead of “find”

We thank the reviewer for pointing this oversight. We have now made the appropriate changes “leads” (line 101) and “found” (line 453).

Line 277: explain what SREBP is.

We have now explained SREBP (line 92).

Reviewer 2:

Summary:

Ng et al. revisited the effect of cholesterol depletion on autophagy and showed that cholesterol depletion by either MBCD or statin promoted autophagy initiation. They then investigated the potential role of an evolutionarily conserved family of ER-anchored lipid transfer proteins, GRAMD1s (GRAMD1A/1B/1C), in the regulation of autophagy and found that the depletion of one of the GRAMD1s (GRAMD1C) facilitated starvation-induced autophagy (possibly by modulating autophagy initiation). Using live cell imaging and biochemical assays, they found that a small fraction of GRAMD1C localized to ER-mitochondria contact sites and that GRAMD1C interacted with mitochondria possibly through its GRAM domain. They were not able to show which mitochondrial protein was responsible for the targeting of GRAMD1C to mitochondria. To further investigate the function of GRAMD1C at ER-mitochondria contact sites, the authors depleted GRAMD1C expression in U2OS cells by RNAi or genetic knockout and found that this manipulation lead to accumulation of cholesterol in mitochondria and enhanced mitochondrial oxidative phosphorylation. Based on these results, they concluded that GRAMD1C functions at ER-mitochondria contacts to regulate mitochondrial bioenergetics. They then investigated the relationship between the expression of GRAMD1s (as well as other proteins that contain the GRAM domain without the sterol-harboring StART-like domain, GRAMD2A, GRAMD2B) and the survival of patients suffering from clear cell renal cell carcinoma and reported that there is some correlation between GRAMD1 expression and patient survival.

Overall, this study provides little mechanistic insights into the role of cholesterol in autophagy. It also remains unclear how GRAMD1C affects mitochondrial bioenergetics. Most importantly, the three major pieces of data that the authors presented in this study, namely 1) effects of cholesterol depletion in autophagy, 2) the potential role of GRAMD1C in the regulation of mitochondrial cholesterol levels and mitochondrial oxidative phosphorylation at ER-mitochondria contact sites, and 3) the association between the severity of clear cell renal cell carcinoma and the expression levels of various GRAMD family proteins, are not well connected to support each other. The authors should narrow their focus of the study and perform more in-depth analysis of selected data sets. With the current form, the study remains descriptive and preliminary for publication.

We thank the reviewer for their insightful and useful comments.

Major comments:

How MBCD affects autophagy remains unclear. It has been known that depletion of cholesterol by 1-hour treatment of MBCD induces autophagy (e.g., PMID: 17056010). If it is independent of mTORC1 as the authors claimed in their current study, what is the molecular mechanism that underlies autophagy induction caused by cholesterol depletion? Authors showed increased appearance of EGFP-BATS puncta both in starved and MBCD-treated conditions in Figure 2h; these results suggest that starvation and cholesterol depletion cause autophagy induction with similar mechanisms rather than distinct mechanisms (opposite to what the authors proposed).

We would like to thank the reviewer for bringing our attention to the work done by Jinglei Cheng et al, which we also have cited in the introduction (Reference 15). Their paper indeed shows increased LC3-II upon 1 h MBCD treatment, however the data does not allow for the comparison of autophagy flux between control and cholesterol depleted cells, thus preventing readers from drawing conclusions of increased/decreased autophagy flux. In this study, we show that MBCD treatment leads to increased autophagy flux in U2OS cells as compared to control treated cells already after 30 min of MBCD treatment (Figure 1a-d, Figure 2a-b). Indeed, we cannot rule out a possible role of mTORC1 in cholesterol induced autophagy, but find that cholesterol depletion leads to a faster induction of autophagy compared to starvation. The increased appearance of EGFP-BATS puncta seen both in starved and MBCD-treated conditions indicates that both conditions lead to increased membrane curvature at autophagy initiation sites, but do not necessarily reflect similar mechanisms of induction. Accordingly, we have now changed the chapter title from “Cholesterol depletion facilitates starvation-induced autophagy in an mTORC1 independent manner” to “Cholesterol depletion alters starvation-induced autophagy dynamics”. Moreover, we speculate on a role for mTORC1 and refer to previous publications linking cholesterol levels to mTORC1 regulation (Line 139-140, reference 30-31).

Another important issue is non-selective action of MBCD for cholesterol depletion; MBCD affects the levels of cholesterol in the PM (where the majority of cellular cholesterol is present) as well as lysosomes, and it might have indirectly affected autophagy induction.

We agree that it is difficult to properly identify the organelle of which cholesterol depletion is crucial for autophagy induction. However, the accumulation of EGFP-BATS, a curvature sensing domain of the early autophagy protein ATG14, in structures also containing the early autophagy markers ATG13, ATG16L1, WIPI2 and LC3B suggests increased curvature of ER-associated autophagy initiation membranes. Moreover, the increased transcription of SREBP2 target genes in GRAMD1C depleted cells (Supplementary figure 4f) indicates reduced ER cholesterol levels. We cannot however rule out that cholesterol in other organelles can indirectly affect autophagy induction, and we discuss this in the manuscript. For example, the lysosomal cholesterol has been shown to regulate mTORC1 signaling (reference 30-31).

This study provides limited insights into how GRAMD1C affects autophagy. Previous studies had shown that another member of GRAMD1s, GRAMD1A, regulates autophagy (PMID: 31222192). In this study,

the authors provided some evidence that supports a role for GRAMD1A as a negative regulator of autophagy (Figures 3 and 4).

In the GRAMD1A paper (PMID: 31222192), the authors observed differences in autophagy at different times of knockdown. After 24 hrs of GRAMD1A knockdown, the authors observed decreased autophagy, while after 72 hrs of GRAMD1A knockdown, no inhibition of autophagy was observed. Given that our knockdowns were carried out for 72 hrs, our observations are indeed in agreement with previously reported data. We suspect that other GRAMD1s are able to compensate for the loss of GRAMD1A (possibly also GRAMD1C), which could possibly explain the difference between 24 hrs and 72 hrs knockdown.

They also reported that GRAMD1C potentially localized to ER-mitochondria contacts and regulated mitochondrial cholesterol levels as well as mitochondrial oxidative phosphorylation (Figure 5). However, how the function of GRAMD1C at ER-mitochondria contacts is related to the regulation of autophagy induction was not addressed.

We show by live imaging that GRAMD1C-EGFP staining overlaps with mitochondria (Mitotracker Red) (Figure 5b) and that the EGFP-GRAM domain only of GRAMD1C forms transient interactions with mitochondria (Figure 5c) and can be found in mitochondrial fractions (Figure 5d). Moreover, MS characterization of the GRAMD1C interactome identified several mitochondrial and ER-mitochondria contact site proteins (Figure 5e) that were validated by western blot (Figure 5f). Importantly, we show that the increased autophagic flux seen in GRAMD1C KO cells (GKO) can be rescued with full length GRAMD1C but not with the Δ GRAM mutant form of GRAMD1C (Figure 3f-g). Additionally, we find that depletion of another mitochondrial protein (TSPO) also implicated in cholesterol transport give a similar increase in LC3B lipidation as siGRAMD1C (Supplementary figure 5f-g), indicating that mitochondrial cholesterol transport regulates autophagy. We propose that the reduced level of cholesterol at ER-mitochondria contact sites promotes increased membrane curvature, leading to recruitment of the curvature-sensitive proteins of the early autophagy initiation machinery.

The full length GRAMD1C was also able to rescue the respiration phenotypes observed in GKO cells (Figure 6g-h). Also, the Δ GRAM mutant form of GRAMD1C is also able to rescue the phenotype, suggesting that the GRAM domain is dispensable for its effects on mitochondrial respiration. Indeed, we find through CO-IP MS studies (Figure 5e) and western blot analysis (Figure 5f) that the Δ GRAM mutant is still able to interact with mitochondrial proteins. We suspect that this is due to the ability of the GRAM proteins to form a complex (PMID: 31724953), and that the GRAM domain of other GRAMD1s is able to compensate for the loss of the GRAM domain of the Δ GRAM mutant. For example, GRAMD1B was previously shown to interact with mitochondria (PMID: 32738348).

Related to the above-mentioned point, the model/conclusion for the function of GRAMD1C proposed by the authors is not well supported. The authors stated in the line 231 “these results indicate that GRAMD1C regulates autophagosome biogenesis by removal of cholesterol from ER membranes”.

However, there is no data showing that GRAMD1C removes cholesterol from ER membranes in this study.

We show by two different methods that cholesterol levels in mitochondria are increased in cells lacking GRAMD1C (Figure 6a-c). As an indirect readout of ER cholesterol abundance, we assessed SREBP2 target gene activation (SREBP2 is activated upon lowering of ER cholesterol). We find that SREBP2 target genes were upregulated in GRAMD1C depleted cells (Supplementary figure 5f), suggesting decreased ER cholesterol.

The authors suggest that GRAMD1C transports cholesterol from mitochondria to the ER at ER-mitochondria contacts, but the evidence supporting this function is weak. More experiments are needed to support a direct role of GRAMD1C in cholesterol transport at ER-mitochondria contacts. Related to this point, how 1) enhanced mitochondrial oxidative phosphorylation (Figure 6d) is caused by 2) elevated levels of cholesterol in mitochondria (Figure 6a-c) in cells lacking GRAMD1C expression is not addressed. It is also unclear how these alterations in mitochondria affect autophagy induction.

We currently lack the tools to measure directionality of inter-organelle cholesterol transport and we are unable to reliably track the movement of organelle specific cholesterol. Nevertheless, in addition to mitochondrial cholesterol measurements (Figure 6a-c), we find through EM that the mitochondria in siGRAMD1C treated cells were poorly stained with OsO₄, a lipid stain that preferably binds to unsaturated fatty acids and can be affected by cholesterol, suggesting altered mitochondrial lipid composition in cells lacking GRAMD1C (Figure 6d).

Additionally, we find that the change in mitochondrial oxidative phosphorylation is not caused by changes in protein levels of the OXPHOS complex (Supplementary figure 5a-b) or mitochondrial morphology (Supplementary figure 4c-e). We find increased SREBP2 target gene expression, which indicates decreased ER cholesterol, but as we do not directly show the cholesterol transport between the ER and mitochondria, we have now made changes to the text to tone down claims of ER-mitochondria cholesterol transport. The title of the manuscript is also changed from “GRAMD1C regulates autophagy initiation and mitochondrial bioenergetics through ER-mitochondrial cholesterol transport” to “GRAMD1C regulates autophagy initiation and mitochondrial bioenergetics”. With regards to how altered mitochondrial cholesterol affects autophagy induction, we find that knockdown of TSPO, a mitochondrial cholesterol transporter leads to increased autophagy (Supplementary figure 5f). In line with our observations in GRAMD1C depleted cells, this data suggests a role for mitochondrial cholesterol in autophagy.

Finally, the reported association between the expression of GRAMDs and clear renal cell carcinoma is interesting but preliminary and largely descriptive, providing limited mechanistic insights and no link to autophagy induction.

We now show that GRAMD1C depletion leads to increased mitochondrial respiration (Figure 7d) and autophagy (Figure 7e) using the renal carcinoma cell lines 786-0 and A489, respectively. Previous

studies have indicated that autophagy contributes to cell growth, migration (PMID: 32717219) and therapeutic resistance (PMID: 32760216) in renal carcinoma, and we have referred these to the discussion (Line 459-461). It would be interesting to carry out future investigations into the involvement of the GRAMs in ccRCC using relevant animal models and specific GRAM protein inhibitors, but that would extend beyond the scope of this manuscript.

No rescue experiments were performed throughout this study, making it hard to determine whether the phenotypes reported in this study were truly due to the loss/reduction of a cholesterol transporting function of GRAMD1C.

To address whether the autophagy and mitochondria respiration phenotypes observed in cells depleted of GRAMD1C (by siRNA and KO), we first generated stable GKO rescue cells expressing GRAMD1C-EGFP, GRAMD1C(Δ GRAM)-EGFP or EGFP-GRAM. Neither of the rescue constructs were able to rescue the autophagy or mitochondrial phenotypes observed in GKO cells, indicating that the EGFP tag might prevent the function of GRAMD1C. We therefore generated untagged GRAMD1C lenti constructs that were transduced in GKO cells to generate stable rescue cell lines. Indeed, the full length GRAMD1C was able to rescue the autophagy and respiration phenotypes observed in GKO cells (Figure 3f-g and Figure 6g-h). Importantly, the GRAM domain was found to be important for rescue of the autophagy phenotype, as GKO cells expressing GRAMD1C(Δ GRAM) demonstrated increased LC3B lipidation (as seen in GKO cells) (Figure 3f-g).

Moreover, both GRAMD1C and GRAMD1C(Δ GRAM) were able to rescue the increased ATP-linked respiration seen in GKO cells (Figure 6g-h), indicating that GRAMD1C indeed regulates mitochondrial respiration. Interestingly, we also find that GKO cells overexpressing the Δ GRAM mutant has decreased ATP-production linked respiration. We suspect that this is caused by the ability of the Δ GRAM mutant to interact with other members of the GRAM family (Figure 5e, Table 1), which could, through the GRAM domain of other GRAM proteins, compensate for the lack of the GRAM domain in the Δ GRAM mutant.

The localization analysis of GRAMD1C (Figure 5) is premature and their statement in line 262 “Thus, our results show that GRAMD1C interacts with the mitochondria through its GRAM domain” is not well supported. They should at least show GRAMD1C lacking the GRAM domain does not interact with mitochondria. It is also unclear which one of the mitochondrial proteins is directly involved in recruiting GRAMD1C to mitochondria (Figure 6e, 6g). The authors should verify the mass spec results with Western Blotting for at least some of the proteins that they identified through IP-MS analysis. They did not find GRAMD1C in anti-EGFP immunoprecipitates from cells expressing GRAMD1C lacking the GRAM domain (i.e., GRAMD1C Δ GRAM) (Table 1b), which needs to be explained or revisited.

We have now repeated the co-IP Mass Spec analysis of GRAMD1C and GRAMD1C Δ GRAM (Figure 5e, table 1) (four replicates of each), and have confirmed several hits by western blot analysis. The yeast orthologue of GRAMD1C, Lam6 was previously shown to interact with mitochondria through its binding

to Tom70 (PMID:26119743). Interestingly, TOMM70A appeared as a top hit in the GRAMD1C interactome, suggesting a similar mechanism of interaction. While we were not able to validate the interaction of full length GRAMD1C with TOMM70A, we were able to show an interaction between the GRAM domain and TOMM70A (Figure 5f). Moreover, we were able to show binding of the GRAM domain and GRAMD1C to the mitochondrial protein RHOT2 (also known as MIRO2) and to the ER-mitochondria contact site marker ACSL4 (Figure 5f). Thus, our data and the previously reported Lam6-Tom70 interaction (PMID: 26119743), indicate that the localization of GRAMD1C to mitochondria is facilitated by its binding to proteins, but we cannot rule out that binding of the GRAM domain to lipids also regulates its mitochondrial association.

It is notable that the Δ GRAM mutant (now detected in the MS analysis) is also able to associate with many of the proteins found in the immunoprecipitates of full length GRAMD1C (Table 1). As the Δ GRAM mutant is capable of associating with other members of the GRAM family, we suspect that it is able to interact to mitochondria through the formation of a GRAM complex, where it associates with mitochondria through other GRAMs. Indeed, GRAMD1B was previously shown to carry a mitochondrial localization signal on its N-terminal (PMID: 32738348). Nevertheless, as we find that the GRAM domain alone is able to interact with RHOT2 (Figure 5f), it is highly probable that the GRAM domain contributes to the localization of GRAMD1C to the mitochondria.

It is unclear why the abundance of cholesterol-associated proteins (STARD9, ERLIN etc.) is altered in GRAMD1C depleted cells (line 278-279).

We now include results from the mass spec analysis showing that GRAMD1C interacts with cholesterol synthesis proteins (SQLE, DHCR24 and DHCR7) (Table 1) and discuss the possibility of direct cholesterol transport immediately after synthesis (Line 272-273). With regards to the cholesterol-associated proteins (STARD9, ERLIN etc.), the sentence has now been changed to indicate that the implications of their change in GRAMD1C depleted cells are not fully known, but it suggests altered cholesterol metabolism (Line 308).

Reviewer 3

In their manuscript, Matthew Yoke Wui Ng and colleagues report that the ER-located protein GRAMD1C specialized in cholesterol transport is a negative regulator of autophagosome biogenesis. This elegant study is based on robust cell biology and biochemistry readouts and deciphers the molecular insights of GRAMD1C function at ER/mitochondria membranes interface. This work shed light on the functional importance of low-cholesterol membrane compartments in autophagy early step regulation. The goal of the study is clearly stated, the manuscript is well written and data are well presented and explained. The physiological relevance brought by the implication of GRAM family proteins in clear cell renal carcinoma (CCR) is of interest, although the direct link of the cellular hallmarks of CCR with autophagosome biogenesis is not very explored. With a dedicated revision, this paper could be of interest for readers of Nature Communications.

We thank the reviewer for these encouraging comments.

I have several remarks concerning the experimental data as well as the general scientific context:

Concerning the mechanistic explanation of low-cholesterol membrane requirement in autophagosome biogenesis: i) have authors checked for ATG14L and VPS34 behavior in low cholesterol cells and/or in GRAMD1C depleted cells? ii) What happens to ER-PI3P positive membranes associated in low cholesterol cells and/or in GRAMD1C depleted cells?

We have used EGFP-BATS, a curvature sensing domain of ATG14, to assess effects of low cholesterol levels and depletion of GRAMD1C on membranes involved in autophagosome biogenesis (Figure 2h-i, Figure 4h-i). We have also stained cells with antibodies recognizing endogenous ATG13, ATG16L1, WIPI2 (a PI3P effector protein) and LC3B (Figure 1g-i, Figure 4a-g), all showing increased puncta formation under these conditions.

In order to investigate VPS34 activity, we investigated EEA1, a PI3P binding protein, as an indirect readout of VPS34 activity. We treated cells \pm 5 μ M VPS34i-IN1 (VPS34 inhibitor, PMID: 25177796) for 1 hr before fixation and immunostaining for EEA1. >500 cells per condition were analyzed. We find that treatment of MBCD leads to a decrease in EEA1 puncta, suggesting decreased PI3P. However, this reduction can most likely be attributed to perturbations of clathrin coated vesicle formation (PMID: 1019805). In both control and MBCD-treated cells, treatment with VPS34-IN1 leads to a reduction of EEA1 puncta. (Figure a included for reviewer below).

While we were not able to assess ER specific PI3P, we investigated global endogenous PI3P using an anti-PI3P antibody. Cells were treated with MBCD \pm 5 μ M VPS34-IN1 (Figure b included for reviewer below) before fixation and immunostaining for PI3P (#Z-P003, Echelon Biosciences). >500 cells per condition were analyzed. Treatment with MBCD did not alter global PI3P intensity. As treatment with VPS34-IN1 did also not alter PI3P staining intensity (although it did affect EEA1 puncta), it is highly likely that the PI3P staining in these experiments were not specific and that the signal observed does not represent PI3P.

We have also cloned ATG14L from U2OS cDNA, and generated EGFP-ATG14L stably expressing U2OS cells. We have imaged and quantified ATG14L puncta in MBCD treated cells (Figure c included for reviewer below). >500 cells per condition were analyzed. While we observe EGFP-ATG14L puncta in cells, these do not increase in number upon amino acid starvation. We also occasionally observed the presence of larger EGFP-ATG14L structures, which we suspect are non-autophagy related aggregates (Figure d included for reviewer below).. These structures lead to inaccurate puncta counting and this experiment will need to be optimized to control the expression level of EGFP-ATG14L. Thus, these results are unfortunately not sufficient to describe ER-specific PI3P in cholesterol depleted cells. However, we show that membrane recruitment of the PI3P-binding protein WIPI2 increases upon GRAMD1C depletion (Figure 4a,d), indicating that VPS34 activity and PI3P formation remains functional in GRAMD1C knockdown cells.

- a. U2OS cells were treated with MBCD ± 5 μM VPS34-IN1 for 1 hr. The cells are stained for EEA1 and imaged. Error Bar = SD. Scale bar = 50 μm.
- b. U2OS cells were treated with MBCD ± 5 μM VPS34-IN1 for 1 hr. The cells are stained for PI3P and imaged. Error Bar = SD. Scale bar = 50 μm
- c. U2OS cells stably expressing EGFP-ATG14L were amino acid starved or treated with MBCD for 1 hr before fixation and imaging. Error Bar = SD. Scale bar = 20μm.
- d. EGFP-ATG14L aggregates. Scale bar = 50 μm.

Concerning the involvement of GRAM family proteins in CCR: in the light of authors' data about GRAMD1C implication in autophagy regulation, it is worthy to notice that autophagic machinery has been linked to proximal tubule renal epithelial cells differentiation and (mitochondria and lipid) metabolism via primary-cilium and mechanical/shear stress. The authors mentioned that starvation-induced autophagy is negatively regulated by GRAMD1C, but what about shear-stress-induced autophagy?

It would be interesting to investigate the role for GRAMD1C in sheer stress-induced autophagy, but we unfortunately lack the setup required for the generation of laminar flow on our cells. We have however included in our discussion a possibility for sheer stress-induced autophagy (Line 457-459) and referred to a number of relevant studies.

The manuscript could benefit from electron microscopy analyzes of "autophagic structures" generated in low cholesterol cells and in GRAMD1C depleted cells, notably concerning the membrane curvature part. In the same line, it could be interesting to decipher whether the boosted set of autophagosomes formed in absence of GRAMD1C are fully functional.

Autophagosomes in cells treated with MBCD were previously observed by EM (PMID:17056010), where the authors indicated that autophagosomes were similar to autophagosomes observed in amino acid starved cells. In order to study autophagosome structure in GRAMD1C depleted cells, we carried out EM analysis of control (siNC) and siGRAMD1C treated cells. We analyzed more than 30 cells, but were not successful in observing autophagosomes in siNC treated cells, and only 4 autophagosomes in siGRAMD1C treated cells (Figure included for reviewer below). As we carried out the experiments under normal conditions (not starved), the rate of autophagosome formation could be improved in the future by amino acid starvation prior to EM analysis. Nevertheless, we do not find obvious defects in the structure of the few autophagosomes observed in the siGRAMD1C treated cells. The increased degradation of long-lived proteins in cells depleted of GRAMD1C (using two different siRNA oligos) (Figure 3c) also suggests that the autophagosomes are functional.

Interestingly, the EM experiment revealed that the membranes and organelles of siGRAMD1C treated cells were weakly stained with OsO₄, a lipid stain that preferably binds to unsaturated fatty acids and can be affected by cholesterol, suggesting altered mitochondrial lipid composition in cells lacking GRAMD1C (Figure 6d).

In the mitochondria fractionation experiments, data could benefit from using empty GFP as a negative control.

We have now included a figure (Figure 5d) showing that free GFP is not found in the mitochondria fraction, but rather is found in the cytosolic fraction (experiments done in parallel to EGFP-GRAM in same figure). The same result (no EGFP in the crude mitochondrial fraction) is shown in another experiment using fractionation of cells expressing EGFP, GRAMD1C-EGFP or Δ GRAM-EGFP (Supplementary figure 4g).

The implication of GRAM/ER-to-plasma membrane cholesterol transfer putative relationship with ER/plasma membrane contact sites (which are also implicated in autophagosome biogenesis) could be discussed.

We have now included a short discussion regarding the possible roles of ER/PM cholesterol transport in autophagy (Line 424). Indeed, the main function of the GRAMs are in PM-to-ER cholesterol transfer, and given earlier reports of PM involvement in autophagosome biogenesis (PMID: 30220461, 28550152), we are not able to rule out a role for ER-PM cholesterol transport in regulation of autophagy. Accordingly, we have now changed the title of the manuscript from “GRAMD1C regulates autophagy initiation and mitochondrial bioenergetics through ER-mitochondria cholesterol transfer” to “GRAMD1C regulates autophagy initiation and mitochondrial bioenergetics”.

It is not clear to me whether “ER-mitochondria contact sites based cholesterol transport regulation” enables direct and local ER membrane lowering of cholesterol, for example at omegasome (other membranes normally positive for cholesterol (such as Golgi or endosomes) have been reported to participate to autophagosome biogenesis as well).

We find ER-mitochondria contact site proteins such as ACSL4, VDAC2, TMX1 in the MS analysis of immunoprecipitates from GRAMD1C (Table 1), indicating that GRAMD1C is also localized to the region between ER and mitochondria. Previous studies suggest that GRAMD1C depletion leads to reduced cholesterol in the ER (PMID: 31724953). Indeed, we do show by two different methods that cholesterol levels in mitochondria are increased in cells lacking GRAMD1C (Figure 6a-c). As an indirect readout of ER cholesterol abundance, we assessed SREBP2 target gene activation (as SREBP2 is activated upon lowering of ER cholesterol). We find that SREBP2 target genes were upregulated in GRAMD1C depleted cells (Supplementary figure 5f), suggesting decreased ER cholesterol. However, as we currently lack the tools for live cell imaging of cholesterol poor membranes such as the ER and mitochondria, we are unable to investigate if the cholesterol transport is local and direct. Indeed, we find that GRAMD1C associates with proteins from different organelles (Table 1), which opens to the possibility for the cholesterol of other organelles such as the Golgi, PM and endosomes in autophagy. We have now included this in the discussion. (Line 422-425).

Could authors discuss a bit more about the absence of P62 readouts associated phenotypes in their study?

p62 plays a central role in selective autophagy such as mitochondria and protein aggregates through its binding to ATG8 proteins and ubiquitin. As such, the lack of p62 flux change in cholesterol/GRAMD1C depleted cells suggest that GRAMD1C is not involved in selective autophagy (in line with our data showing no effect on Parkin-dependent or -independent mitophagy), but is rather involved in canonical autophagy. Additionally, p62 is altered by cell stress such as oxidative stress (PMID: 26117325), proteasome inhibition (PMID: 23922739), and p62 is quickly replenished during starvation (PMID: 24394643), which might also contribute to the lack of change in p62 turnover in our experiments.

Could authors specify the use of U2OS cells in the context of the study?

Our lab previously generated U2OS cells stably expressing mitophagy and autophagy reporters (PMID: 34671015) and have optimized protocol for these cells. The cells were initially chosen due to their flat morphology which leads to less out of focus noise during imaging. Nevertheless, we successfully replicated the increased autophagy phenotype resulting from GRAMD1C depletion in A498 cells (Figure 7e) and the increased ATP-production linked respiration in 786-O cells (Figure 7d), suggesting that the observed phenotypes are not a cell line specific observation.

I was wondering why western blots of LC3 only displayed LC3-II form?

LC3-I is also stained by the LC3 antibody. However, the intensity of the LC3-I is much weaker compared to the LC3-II form and is only visible upon high exposure (see examples in figure included for reviewer only below). Thus, in some of the blots, the LC3-I band is hardly visible. Nevertheless, because the quantification of the bands is done by quantification of LC3-II levels relative to Actin (as recommended by the guidelines for interpretation of autophagy assays), we found it sufficient to represent only the autophagosome associated LC3-II.

2 examples of high and low exposure LC3B blots.

At low exposure, LC3B-I bands are less visible.

At high exposure, LC3B-I bands become visible, but LC3B-II bands become oversaturated.

Reference number 3 is not referring to fatty acids enrichment in autophagosomes membranes.

In reference number 3, Hajnalka et al. found through lipidomic studies of autophagosomes isolated from *Drosophila* that autophagosomes are composed of about 63.5% unsaturated fatty acids. We have now included reference to an additional study that also indicate that autophagosomes from yeast cells have a high composition of unsaturated fatty acids (PMID: 31883797).

Reviewer #1 (Remarks to the Author):

The authors adequately resolved all of the critical points that I raised - either by revising the manuscript or by performing additional experiments that corroborated their initial data. Therefore, I am happy to recommend this study for publication. Well done!

Reviewer #2 (Remarks to the Author):

Summary:

Although the authors addressed some of my concerns with new data sets and text changes, the major issue of this manuscript is unresolved; the three pieces of their findings do not support each other well (and, in cases, contradict with each other) [1) effects of cholesterol depletion in autophagy, 2) the potential role of GRAMD1C in the regulation of mitochondrial cholesterol levels and mitochondrial oxidative phosphorylation at ER-mitochondria contact sites, and 3) the association between the severity of clear cell renal cell carcinoma and the expression levels of various GRAMD family proteins]. The current manuscript remains descriptive, possibly presenting unrelated data sets.

Major issues of the current manuscript:

1. The conclusion in their model figure is not well supported by the data. In view of the newly provided data sets, whether GRAMD1C acts at ER-mitochondria contacts vs. other contacts (e.g., ER-plasma membrane contacts) to control autophagy dynamics needs to be shown with additional experiments.

With their new experiments, the authors showed that increased autophagy flux phenotype in GRAMD1C knock-out (GKO) cells was rescued by re-expression of GRAMD1C full-length, but not by expression of GRAMD1C lacking the GRAM domain (Δ GRAM) (Figure 3g). However, increased mitochondrial bioenergetics phenotype in GKO cells was rescued by expression of either one of GRAMD1C full-length or Δ GRAM (Figure 6h). These new results suggest that GRAMD1C controls autophagy and mitochondrial bioenergetics via distinct mechanisms (the former relies on the GRAM domain, while the latter does not rely on the GRAM domain according to the new data). The authors speculated that the rescue of mitochondrial bioenergetics phenotype with Δ GRAM could be due to its unaffected interaction with mitochondria (line 315) (possibly via oligomerization of Δ GRAM with other GRAM proteins, e.g. Figure 5f). If this is indeed the case, the "inability" of Δ GRAM to rescue the autophagy flux phenotype (Figure 3g) is not due to the loss of GRAMD1C function at ER-mitochondria contacts, but to the loss of its function at other contacts (because Δ GRAM can still localize to ER-mitochondria contacts).

These results raise major concerns in the interpretation of the data in this manuscript; it may not be the function of GRAMD1C at ER-mitochondria contacts but the function of GRAMD1C at other contact sites, such as ER-plasma membrane contacts, which may contribute to the regulation of autophagy induction. In this case, the characterization of mitochondrial phenotype in GRAMD1C-depleted cells would be irrelevant to this study. In other words, the changes in mitochondrial bioenergetics or mitochondrial cholesterol levels observed in GRAMD1C-depleted cells may not be related to (or causative to) the changes in the dynamics of autophagy in these cells.

Increased SREBP2 target gene expression in GRAMD1C-depleted cells shown by the authors (Supplementary Figure 4f) could be due to reduced PM to ER cholesterol transport in GRAMD1C-depleted cells rather than defects in cholesterol transport between mitochondria and the ER.

This issue needs to be addressed with additional experimentation as the novel aspect of this study was a potential role of GRAMD1C at ER-mitochondria contacts in autophagy regulation.

2. The new TSPO knock-down experiment/result does not support the model where mitochondrial cholesterol accumulation in GRAMD1C-depleted cells is related to increased autophagy induction.

In the revised manuscript, the authors performed TSPO knock-down experiments with the aim of linking mitochondrial cholesterol levels with the degree of autophagy induction. TSPO is generally considered as an importer of cholesterol to mitochondria (PMID: 21119734), whose knock-down is expected to “decrease” the amount of cholesterol in mitochondria. This is opposite to what the authors showed with GRAMD1C depletion, which supposedly “increase” mitochondria cholesterol levels. Yet, both GRAMD1C depletion and TSPO depletion cause increase in autophagy flux (Figure 3 vs. Supplementary Figure 5f). Thus, this experiment shows the cholesterol transporting function of GRAMD1C at ER-mitochondria contacts may not be related to the regulation of autophagy.

Related to this point, the data which showed accumulation of cholesterol in mitochondria with mCherry-D4 are not convincing. The authors did not address my previous concern on the usage of mCherry-D4. As mentioned previously, there is very little reduction of D4 binding to mitochondria upon MBCD treatment in both wild-type and GKO cells (Figure 6a), contrary to their statement in line 285 (the authors claim that mCherry-D4 selectively binds to cholesterol on isolated mitochondria; instead their data show that large fraction of mCherry-D4 remains present in MBCD-treated mitochondria). The extent of the reduction of mCherry-D4 binding to mitochondria looks even more pronounced in wild-type cells compared to GKO cells, indicating that the cholesterol levels are actually decreased in mitochondria upon GRAMD1C depletion. This is opposite to the colorimetric quantification of cholesterol shown in Figure 6. More quantitative measurement of cholesterol (e.g., mass spec analysis) should be performed instead of the D4-based assays because D4 binding can be influenced by other factors such as accessibility of cholesterol and phospholipid compositions (e.g., PMID: 25809258; PMID: 23754385).

3. The association of GRAMDs and clear cell renal carcinoma (ccRCC) remains descriptive with no any linkage shown for autophagy nor mitochondria bioenergetics.

The association of the levels of GRAMD1C expression with overall survival rates in ccRCC patients may not be relevant to GRAMD1C's role in the regulation of autophagy or mitochondria bioenergetics for the following reasons:

a) In the 786-O ccRCC cells treated with siGRAMD1C, neither relative colony area (Figure 7c) nor wound healing (Supplementary Figure 7a, b) was affected. In these cells, KI67 positive proliferative cells were reduced (Figure 7e). These results do not support the association of low GRAMD1C expression with reduction in overall survival rate in ccRCC patients (Figure 7a).

b) In Figure 3, the authors systematically characterized the effects of RNAi knockdown of GRAMD1A, GRAMD1B, GRAMD1C, GRAMD2, and GRAMD3 in autophagy and showed that GRAMD1C knock-down had most impact on the starvation-induced autophagy. However, the data presented in Figure 7a shows that the expression of all GRAM proteins are associated with some changes in overall survival rates of ccRCC patients. In fact, siGRAMD1A and siGRAMD1B showed highest reduction in the relative colony area in the 786-O ccRCC cells (Figure 7c) despite their little impact on autophagy (Figure 3). The association of the various GRAM proteins and in overall survival rates of ccRCC patients shown in Figure 7a is not supported by any of the data shown in other figures.

The expression of GRAM proteins may be correlated with overall survival rates of ccRCC patients one way or another due to their function that is not necessarily related to the regulation of autophagy induction or mitochondrial bioenergetics. Showing that depletion of GRAMD1C in A498 cells or 786-O ccRCC cells causes changes in autophagy induction or mitochondrial bioenergetics (similar to depletion of GRAMD1C in U2OS cell) does not support the relevance of the association of GRAMD1C expression in the overall survival rate of ccRCC patients.

Other relevant issues:

1. It remains unclear whether cholesterol-transporting function of GRAMD1C is related to the observed phenotypes, including changes in mitochondrial bioenergetics or changes in autophagy induction. The authors should use a version of GRAMD1C lacking the VAS_t domain or possessing the mutated VAS_t domain that inhibits cholesterol transport (e.g., PMID: 31724953) (cholesterol

transport-defective mutant), along with GRAMD1C full-length and Δ GRAM, in their various rescue experiments to see if any of the phenotypes described in this study is related to cholesterol transporting function of GRAMD1C or not. Figure 6a-c could be more convincing if they are presented along with those rescue experiments to help support the conclusion (line 390-391).

2. The rescue experiments with untagged GRAMD1C are good addition in the revised manuscript. However, the authors also found that C-terminally tagged GRAMD1C (GRAMD1C-EGFP) is not functional. This is a major concern for the localization analysis (Figure 5b, Supplementary Figure 3a and others) as well as for the overexpression experiments (Supplementary Figure 3c,d). These experiments should be replaced by functional GRAMD1C proteins (e.g., N-terminally tagged EGFP-GRAMD1C etc.).

3. Discuss the effects of MBCD in more detail (e.g., PMID: 17493580). It reduces cholesterol levels of the plasma membrane (as MBCD does not enter cells), but depending on the amount and duration of the treatment, MBCD can affect cholesterol levels of various cellular membranes. Because the authors claim the significance/novelty of short-term MBCD treatment in their study, they should consider measuring the levels of cholesterol in various cellular membranes (e.g., total membranes, mitochondria, the ER etc.) upon the short-term MBCD treatment. This will allow the authors to see how cholesterol levels are affected in various compartments by the short-term MBCD treatment (for example, does it affect mitochondria's cholesterol levels or ER's cholesterol levels or both?) and how such effects are correlated with facilitation of autophagy (e.g., 30 min treatment with MBCD or ATV). Changes in cholesterol levels in the ER maybe the root cause of the changes in autophagy induction in both MBCD treated cells and GRAMD1C depleted cells (rather than GRAMD1C's function at ER-mitochondria contacts; see above for the comments on SREBP2 target gene expression).

4. Andersen et al., previously showed that GRAMD1B/Aster-B transports cholesterol from the ER to mitochondria and that depletion of GRAMD1B resulted in decrease in cholesterol levels of mitochondria and decrease in mitochondrial bioenergetics (PMID: 32738348). GRAMD1B is a paralog of GRAMD1C. How do the authors explain such discrepancy?

5. siRNA against GRAMD1B, GRAMD2A, and GRAMD2B is not so effective according to their new data (Supplementary Figure 1e). Can the authors use CRISPR/Cas9-mediated knock-out approach or other siRNA targets to confirm that depletion of other GRAM proteins does not affect autophagy? Otherwise, it is unclear how to interpret the data in Figure 3a.

Minor points:

Line 116: it would be good to explain why you add BafA1 for broader audience.

Supplementary Figure 3d-e (Line 200) comes earlier than Supplementary Figure 3a (Line 223) and Supplementary Figure 3b-c (Line 231).

Line 227: ..GRAMD1C regulates autophagosome initiation by promoting increased membrane curvature >> should it be "..GRAMD1C inhibits autophagosome initiation by reducing membrane curvature" ?

Line 232: GRAMD1C regulates autophagosome biogenesis through increased recruitment of early autophagic markers >> should it be "GRAMD1C inhibits autophagosome biogenesis through suppression of the recruitment of early autophagic markers" ?

Line 256: As expected >> cite Naito et al., PMID: 31724953

Line 295: citation 43 does not refer to this finding.

Supplementary Figure 4g: it would be helpful to indicate which bands correspond to the target proteins (but refer to the point #2 in the section of "other relevant issues")

Line 660: Related to the cholesterol quantification method, was it performed in cell pellets or

mitochondria? Please clarify.

Reviewer #3 (Remarks to the Author):

The revised version replies to most of my comments and improves the manuscript quality.

Rebuttal: “GRAMD1C regulates autophagy initiation and mitochondrial bioenergetics” (NCOMMS-21-25735A).

Reviewer #1 (Remarks to the Author):

The authors adequately resolved all of the critical points that I raised - either by revising the manuscript or by performing additional experiments that corroborated their initial data. Therefore, I am happy to recommend this study for publication. Well done!

We thank the reviewer for taking the time to review our manuscript and for the constructive comments that significantly improved the manuscript.

Reviewer #2 (Remarks to the Author):

1. The conclusion in their model figure is not well supported by the data. In view of the newly provided data sets, whether GRAMD1C acts at ER-mitochondria contacts vs. other contacts (e.g., ER-plasma membrane contacts) to control autophagy dynamics needs to be shown with additional experiments.

With their new experiments, the authors showed that increased autophagy flux phenotype in GRAMD1C knock-out (GKO) cells was rescued by re-expression of GRAMD1C full-length, but not by expression of GRAMD1C lacking the GRAM domain (Δ GRAM) (Figure 3g). However, increased mitochondrial bioenergetics phenotype in GKO cells was rescued by expression of either one of GRAMD1C full-length or Δ GRAM (Figure 6h). These new results suggest that GRAMD1C controls autophagy and mitochondrial bioenergetics via distinct mechanisms (the former relies on the GRAM domain, while the latter does not rely on the GRAM domain according to the new data). The authors speculated that the rescue of mitochondrial bioenergetics phenotype with Δ GRAM could be due to its unaffected interaction with mitochondria (line 315) (possibly via oligomerization of Δ GRAM with other GRAM proteins, e.g. Figure 5f). If this is indeed the case, the “inability” of Δ GRAM to rescue the autophagy flux phenotype (Figure 3g) is not due to the loss of GRAMD1C function at ER-mitochondria contacts, but to the loss of its function at other contacts (because Δ GRAM can still localize to ER-mitochondria contacts).

These results raise major concerns in the interpretation of the data in this manuscript; it may not be the function of GRAMD1C at ER-mitochondria contacts but the function of GRAMD1C at other contact sites, such as ER-plasma membrane contacts, which may contribute to the regulation of autophagy induction. In this case, the characterization of mitochondrial phenotype in GRAMD1C-depleted cells would be irrelevant to this study. In other words, the changes in mitochondrial bioenergetics or mitochondrial cholesterol levels observed in GRAMD1C-depleted cells may not be related to (or causative to) the changes in the dynamics of autophagy in these cells.

Increased SREBP2 target gene expression in GRAMD1C-depleted cells shown by the authors (Supplementary Figure 4f) could be due to reduced PM to ER cholesterol transport in GRAMD1C-

depleted cells rather than defects in cholesterol transport between mitochondria and the ER.

This issue needs to be addressed with additional experimentation as the novel aspect of this study was a potential role of GRAMD1C at ER-mitochondria contacts in autophagy regulation.

We agree with the reviewer that we cannot rule out the possibility that the regulation of autophagy by GRAMD1C is caused by cholesterol movement between the ER and the PM (or other organelles). It is possible that GRAMD1C can interact with several organelles at the same time (as seen for the yeast ortholog Lam6). Indeed, we do not claim that the role of GRAMD1C as a negative regulator of autophagy is solely due to ER-mitochondrial cholesterol transport, as already discussed in our manuscript (line 415-422). However, it is well recognized in the autophagy field that autophagosomes initiate from ER-mitochondria contact sites (called omegasomes) and it was therefore interesting for us to further characterize the role of GRAMD1C at such contact sites.

In response to this reviewer's comment, we have also made changes to the title of the previous version of this manuscript from "GRAMD1C regulates autophagy initiation and mitochondrial bioenergetics through ER-mitochondrial cholesterol transport" to "The cholesterol transport protein GRAMD1C regulates autophagy initiation and mitochondrial bioenergetics". Additionally, we have now made changes to the graphical abstract to reflect the possible involvement of the PM (Supplementary figure 7c).

Our current data indicate that GRAMD1C regulates autophagy and mitochondrial cholesterol transport, and we refrain from stating that GRAMD1C regulates autophagy through mitochondrial cholesterol transport.

Referring to the comment "*If this is indeed the case, the "inability" of Δ GRAM to rescue the autophagy flux phenotype (Figure 3g) is not due to the loss of GRAMD1C function at ER-mitochondria contacts, but to the loss of its function at other contacts (because Δ GRAM can still localize to ER-mitochondria contacts).*" We agree with the reviewer that a general decrease in ER cholesterol level in GRAMD1C depleted cells likely is the main contributor to the autophagy phenotype, rather than ER-Mitochondria cholesterol transport only. The loss of the GRAM domain would prevent GRAMD1C from interacting with the PM (PMID: 30220461), and thus reduce ER cholesterol import. The results shown in Supplementary figure 2a-b demonstrate that the GRAM domain of GRAMD1C is required for its role in autophagy and we do show by several experimental approaches that the GRAM domain interacts with mitochondria (Figure 5c,d and f). The GRAM domain is in essence a PH like lipid binding domain, which can possibly bind to cholesterol rich microdomains at the PM or MAMs during autophagy initiation. It was previously reported to interact with different lipid species (PI3P, PI4P, PI5P, PI(3,5)P₂) that are found in different cellular compartments (PMID: 3122192). Because of its ability to interact with many different cellular compartments and lipids, we cannot rule out a possibility of other organellar contact sites in the autophagy flux phenotype. Moreover, as the GRAM family proteins have been shown to oligomerize (PMID: 31724953) and GRAMD1B can transport cholesterol from the ER to mitochondria (PMID: 32738348) it is also possible that they are able to compensate for the loss of the GRAMD1C GRAM domain in the case of the mitochondrial

bioenergetics phenotype. However, in response to this reviewer's comment, we have now generated GRAMD1C KO cells rescued with GRAMD1C lacking the VAS_t domain and show that this domain, and by extension GRAMD1C-mediated cholesterol transport, is required for the role of GRAMD1C both in autophagy and mitochondrial bioenergetics (Figure 3f-g, Supplementary figure 5a-b).

2. The new TSPO knock-down experiment/result does not support the model where mitochondrial cholesterol accumulation in GRAMD1C-depleted cells is related to increased autophagy induction.

In the revised manuscript, the authors performed TSPO knock-down experiments with the aim of linking mitochondrial cholesterol levels with the degree of autophagy induction. TSPO is generally considered as an importer of cholesterol to mitochondria (PMID: 21119734), whose knock-down is expected to "decrease" the amount of cholesterol in mitochondria. This is opposite to what the authors showed with GRAMD1C depletion, which supposedly "increase" mitochondria cholesterol levels. Yet, both GRAMD1C depletion and TSPO depletion cause increase in autophagy flux (Figure 3 vs. Supplementary figure 5h-i). Thus, this experiment shows the cholesterol transporting function of GRAMD1C at ER-mitochondria contacts may not be related to the regulation of autophagy.

We would like to thank the reviewer for the feedback regarding the TSPO experiment. TSPO (also known as PBR) is an integral component of a complex that facilitates cholesterol transport from the OMM to the IMM (PMID: 21976778, 22973050, 2168398), and as such it is not correct to assume that TSPO knockdown decreases total mitochondrial cholesterol, as one would expect cholesterol to accumulate on the OMM upon failure to be imported into the IMM. This is however not clear, as the majority of previous studies of TSPO relied on observing the production of steroid hormones (which are derived from cholesterol), a process that occurs in the mitochondrial matrix, as an indirect readout of cholesterol transport into the IMM/matrix.

Nevertheless, TSPO knock down was done to investigate if inhibition of another mitochondrial cholesterol transport protein also caused an autophagy phenotype. Indeed, we find that depletion of TSPO promotes autophagy flux (Supplementary figure 5h-i). We are however not able to attribute this phenotype to inhibited mitochondrial cholesterol transport (or possibly an accumulation of cholesterol in the OMM) and cannot rule out that this phenotype could be caused by changes to mitochondrial ATP production.

Related to this point, the data which showed accumulation of cholesterol in mitochondria with mCherry-D4 are not convincing. The authors did not address my previous concern on the usage of mCherry-D4. As mentioned previously, there is very little reduction of D4 binding to mitochondria upon MBCD treatment in both wild-type and GKO cells (Figure 6a), contrary to their statement in line 285 (the authors claim that mCherry-D4 selectively binds to cholesterol on isolated mitochondria; instead their data show that large fraction of mCherry-D4 remains present in MBCD-treated mitochondria). The extent of the reduction of mCherry-D4 binding to mitochondria looks even more pronounced in wild-type cells compared to GKO cells, indicating that the cholesterol levels are

actually decreased in mitochondria upon GRAMD1C depletion. This is opposite to the colorimetric quantification of cholesterol shown in Figure 6. More quantitative measurement of cholesterol (e.g., mass spec analysis) should be performed instead of the D4-based assays because D4 binding can be influenced by other factors such as accessibility of cholesterol and phospholipid compositions (e.g., PMID: 25809258; PMID: 23754385).

The MBCD protocol used in our study was not sufficient to remove 100% of all cholesterol from mitochondrial membranes, resulting in residual mCherry-D4 binding to the mitochondrial fraction. Indeed, even at very high concentrations of MBCD (10-25 mM) at long time periods (up to 24 hrs) (we used 5 mM, 30 mins), others have not been able to remove all cholesterol from cells (PMID: 22859971, 29615119, 17493580). As such, we do not expect to completely remove all mitochondrial cholesterol using our protocol. The higher amounts of mCherry-D4 on MBCD treated mitochondria from GKO cells (Figure 6a-b) supports our results that there is more cholesterol on mitochondria isolated from GKO cells and is in agreement with the colorimetric experiment in Figure 6c. Nevertheless, in order to validate this assay further, we carried out MBCD treatment of mitochondria for 1 hr at different concentrations at 37°C. We find that extending the treatment time to 1 hr with MBCD greatly reduced mCherry-D4 probe, indicating that the probe binds specifically to cholesterol (Figure included below for reviewer only).

Figure 1: Mitochondria from U2OS cells were isolated and incubated with mCherry-D4 in buffer containing the indicated concentrations of MBCD for 1 hr. The mitochondria are then washed and subjected to western blot analysis.

3. The association of GRAMDs and clear cell renal carcinoma (ccRCC) remains descriptive with no any linkage shown for autophagy nor mitochondria bioenergetics.

The association of the levels of GRAMD1C expression with overall survival rates in ccRCC patients may not be relevant to GRAMD1C's role in the regulation of autophagy or mitochondria bioenergetics for the following reasons:

a) In the 786-O ccRCC cells treated with siGRAMD1C, neither relative colony area (Figure 7c) nor wound healing (Supplementary Figure 7a-b) was affected. In these cells, KI67 positive proliferative cells were reduced (Figure 7e). These results do not support the association of low GRAMD1C expression with reduction in overall survival rate in ccRCC patients (Figure 7a).

b) In Figure 3, the authors systematically characterized the effects of RNAi knockdown of GRAMD1A,

GRAMD1B, GRAMD1C, GRAMD2, and GRAMD3 in autophagy and showed that GRAMD1C knock-down had most impact on the starvation-induced autophagy. However, the data presented in Figure 7a shows that the expression of all GRAM proteins are associated with some changes in overall survival rates of ccRCC patients. In fact, siGRAMD1A and siGRAMD1B showed highest reduction in the relative colony area in the 786-O ccRCC cells (Figure 7C) despite their little impact on autophagy (Figure 3). The association of the various GRAM proteins and in overall survival rates of ccRCC patients shown in Figure 7a is not supported by any of the data shown in other figures.

The expression of GRAM proteins may be correlated with overall survival rates of ccRCC patients one way or another due to their function that is not necessarily related to the regulation of autophagy induction or mitochondrial bioenergetics. Showing that depletion of GRAMD1C in A498 cells or 786-O ccRCC cells causes changes in autophagy induction or mitochondrial bioenergetics (similar to depletion of GRAMD1C in U2OS cell) does not support the relevance of the association of GRAMD1C expression in the overall survival rate of ccRCC patients.

We agree with the reviewer that our results showing a role for GRAMD1C in autophagy and mitochondrial bioenergetic phenotypes in ccRCC cell lines (786-O) are not necessarily directly underlying the survival curves seen for ccRCC patients. Nevertheless, the correlation between GRAM expression levels and patient overall survival suggests a relationship between the GRAM family of cholesterol transport proteins and ccRCC. The mechanisms behind this are most likely more complex than just cell proliferation, changes to mitochondrial respiration or autophagy. Indeed, autophagy plays both pro- and anti-tumor activities in cancer. Furthermore, the survival curves are generated from patients having undergone cancer therapies, adding an additional layer of complexity compared to our experimental ccRCC cell lines. A complete characterization of how autophagy regulates ccRCC is beyond the scope of this manuscript. Nevertheless, in this manuscript we have identified GRAMD1C as regulator of autophagy in ccRCC cell lines (confirming our data in U2OS cells), and we highlight the fact that several members of the GRAM family appear to play a role in ccRCC survival.

Other relevant issues:

1. It remains unclear whether cholesterol-transporting function of GRAMD1C is related to the observed phenotypes, including changes in mitochondrial bioenergetics or changes in autophagy induction. The authors should use a version of GRAMD1C lacking the VAS_t domain or possessing the mutated VAS_t domain that inhibits cholesterol transport (e.g., PMID: 31724953) (cholesterol transport-defective mutant), along with GRAMD1C full-length and Δ GRAM, in their various rescue experiments to see if any of the phenotypes described in this study is related to cholesterol transporting function of GRAMD1C or not. Figure 6a-c could be more convincing if they are presented along with those rescue experiments to help support the conclusion (line 390-391).

We thank the reviewer for this suggestion to investigate the role of the VAS_t domain in autophagy. We have now generated GKO cells expressing GRAMD1C lacking the VAS_t domain (Δ VAS_t mutant)

and analyzed its ability to rescue autophagy and mitochondrial bioenergetics. Indeed, our new data show that the GRAMD1C Δ VAS_t mutant was unable to rescue the autophagy (Figure 3f-g) and mitochondrial respiration (Supplementary figure 5a-b) phenotypes observed in GKO cells, indicating that GRAMD1C-mediated cholesterol transport is involved in regulation of autophagy and mitochondrial bioenergetics.

2. The rescue experiments with untagged GRAMD1C are good addition in the revised manuscript. However, the authors also found that C-terminally tagged GRAMD1C (GRAMD1C-EGFP) is not functional. This is a major concern for the localization analysis (Figure 5b, Supplementary Figure 3a and others) as well as for the overexpression experiments (Supplementary Figure 3 c-d). These experiments should be replaced by functional GRAMD1C proteins (e.g., N-terminally tagged EGFP-GRAMD1C etc.).

We do not believe that C-terminally tagged GRAMD1C (GRAMD1C-EGFP) is non-functional. We show that GRAMD1C-EGFP is localized to the ER, indicating that this tag does not interfere with its integration into ER membranes. The reason for selecting the C-terminal tag is because the GRAM domain is located close to the N-terminal, and in order to avoid interrupting possible interactions of the GRAM domain with membranes/proteins, we tagged GRAMD1C on the C-terminal where the tag will be localized in the ER lumen. As there are no antibodies recognizing endogenous GRAMD1C we have to rely on localization of tagged GRAMD1C constructs for localization studies. However, all our experiments are done in stable cell lines expressing relatively low levels of GRAMD1C.

We initially made GRAMD1C KO cells rescued with EGFP (control), GRAMD1C-EGFP and Δ GRAM-EGFP and carried out autophagy experiments (not included in this manuscript). We were not able to see significant rescue in GKO cells overexpressing GRAMD1C-EGFP compared to GKO cells overexpressing EGFP only. However, as the EGFP was much highly expressed compared to GRAMD1C-EGFP we were uncertain if overexpression of the EGFP tag alone in the GKO cells affected autophagy. We therefore chose to repeat this experiment with untagged GRAMD1C constructs (Supplementary figure 1h) to avoid having to express a control tag in control GKO cells. As shown in the revised manuscript, we find that untagged GRAMD1C rescue both the autophagy (Figure 3f-g and Supplementary figure 2a-b) and the mitochondrial phenotypes (Figure 6g-h and Supplementary figure 5a-b).

3. Discuss the effects of MBCD in more detail (e.g., PMID: 17493580). It reduces cholesterol levels of the plasma membrane (as MBCD does not enter cells), but depending on the amount and duration of the treatment, MBCD can affect cholesterol levels of various cellular membranes. Because the authors claim the significance/novelty of short-term MBCD treatment in their study, they should consider measuring the levels of cholesterol in various cellular membranes (e.g., total membranes, mitochondria, the ER etc.) upon the short-term MBCD treatment. This will allow the authors to see how cholesterol levels are affected in various compartments by the short-term MBCD treatment (for example, does it affect mitochondria's cholesterol levels or ER's cholesterol levels or both?) and how such effects are correlated with facilitation of autophagy (e.g., 30 min treatment with MBCD or ATV). Changes in cholesterol levels in the ER maybe the root cause of the changes in autophagy induction

in both MBCD treated cells and GRAMD1C depleted cells (rather than GRAMD1C's function at ER-mitochondria contacts; see above for the comments on SREBP2 target gene expression).

Similar to the reviewer, we believe that changes to ER cholesterol pools regulates autophagy induction. We have also highlighted the involvement of GRAMD1C in ER-PM cholesterol transport in several parts of the manuscript (line 89-93, 396-398, 435-439). Indeed, our initial hypothesis was that cholesterol removal from ER associated autophagosome initiation sites promote the generation of curved membranes during *de novo* synthesis of autophagic membranes. Supporting this, we show that cholesterol removal (by MBCD) lead to a significant increase of EGFP-BATS puncta (ER associated curvature sensor) compared to control cells (Figure 2h-i).

As suggested by the reviewer, we have carried out a cell fractionation experiment to measure total cellular, ER, and mitochondrial cholesterol in cells upon short-term MBCD treatment. ER isolation was carried out using the ER isolation kit (Sigma, ER0100), crude mitochondria purification was carried out as described in the manuscript and cholesterol quantification was carried out using a cholesterol quantification kit (Abcam, ab65359). We find that MBCD treatment for 1 hr leads to a dose dependent decrease of cholesterol of all membranes, with the largest effects on total cell cholesterol and ER cholesterol (Figure included below for reviewer only).

Figure 2: U2OS cells were incubated in the indicated concentration of MBCD for 1 hr. Crude mitochondria and ER are isolated from these cells and total cholesterol was measured. Cholesterol abundance were normalized to protein abundance, and are set relative to untreated samples (MBCD 0 mM).

4. Andersen et al., previously showed that GRAMD1B/Aster-B transports cholesterol from the ER to mitochondria and that depletion of GRAMD1B resulted in decrease in cholesterol levels of mitochondria and decrease in mitochondrial bioenergetics (PMID: 32738348). GRAMD1B is a paralog of GRAMD1C. How do the authors explain such discrepancy?

The results from Andersen et al. does not indicate if the measurements of mitochondrial respiration are normalized. With regards to mitochondrial respiration, normalization (using per μg protein or per 1000 cells) is especially crucial as mitochondrial respiration measurements in the Seahorse analyzer are highly sensitive to the number of cells. For example, a well with 20000 cells will theoretically give twice the reading compared to a well with 10000 cells. In our experience, GRAMD1B knockdown often leads to slower cell growth in U2OS cells and 786-O cells, which could contribute to the lower mitochondrial respiration that they observed.

Furthermore, Andersen et al. show that GRAMD1B remains associated with the mitochondria in GRAMD1B lacking the mitochondria localization signal (Δ 2-31) (PMID: 32738348, Figure 4d and 4e). This is quite similar to our observation in the Δ GRAM mutant being able to interact with mitochondria. As the GRAMs have been shown to oligomerize (PMID: 31724953), we believe that they are able to compensate for the loss of the GRAM domain or in the case of GRAMD1B, the loss of the mitochondria localization signal.

5. siRNA against GRAMD1B, GRAMD2A, and GRAMD2B is not so effective according to their new data (Supplementary Figure 1e). Can the authors use CRISPR/Cas9-mediated knock-out approach or other siRNA targets to confirm that depletion of other GRAM proteins does not affect autophagy? Otherwise, it is unclear how to interpret the data in Figure 3a.

We find that siRNA against GRAMD2A and GRAMD2B leads to around 50% and 70% decrease in mRNA levels. While not complete removal of mRNA, we believe that at these levels, 50-70% knockdown should lead to a reduction in their respective proteins. Coupled with the knockdown efficiency data, the readers should be able to interpret figure 3a accordingly. Nevertheless, our study into the autophagy and mitochondrial respiration phenotypes are focused on GRAMD1C, and we find that the knockdown efficiency of siGRAMD1C-1 and siGRAMD1C-2 to be very good.

Minor points:

Line 116: it would be good to explain why you add BafA1 for broader audience.

We have now added a sentence (line 118) to explain the use of BafA1 in studying autophagy flux and included a reference (PMID: 9639028).

Supplementary Figure 3d-e (Line 200) comes earlier than Supplementary Figure 3a (Line 223) and Supplementary Figure 3b-c (Line 231).

The order of these supplementary figures has now been changed.

Line 227: ..GRAMD1C regulates autophagosome initiation by promoting increased membrane curvature >> should it be “..GRAMD1C inhibits autophagosome initiation by reducing membrane curvature” ?

We would like to thank the reviewer for pointing this out. We have now made changes to the text (Line 232).

Line 232: GRAMD1C regulates autophagosome biogenesis through increased recruitment of early autophagic markers >> should it be “GRAMD1C inhibits autophagosome biogenesis through suppression of the recruitment of early autophagic markers” ?

We have now changed the text to read “...GRAMD1C regulates autophagosome biogenesis through suppression of membrane curvature and the recruitment of early autophagic markers to ER associated initiation sites”. (Line 235-237).

Line 256: As expected >> cite Naito et al., PMID: 31724953

The reference has been added as #22 (Line 261).

Line 295: citation 43 does not refer to this finding.

We would like to thank the reviewer for pointing this out. The citation has been changed to Naito et al., PMID: 31724953 (Line 298).

Supplementary Figure 4g: it would be helpful to indicate which bands correspond to the target proteins (but refer to the point #2 in the section of “other relevant issues”)

We have added molecular weight markers on the right side of the western blot to help with the identification of the correct bands.

Line 660: Related to the cholesterol quantification method, was it performed in cell pellets or mitochondria? Please clarify.

Cholesterol quantification was carried out in isolated mitochondria. We have now made changes to the protocol (Line 675).

Reviewer #3 (Remarks to the Author):

The revised version replies to most of my comments and improves the manuscript quality.

We thank the reviewer for taking the time to review our manuscript and for the constructive comments that significantly improved the manuscript.

Reviewer #2 (Remarks to the Author):

The authors addressed my major concerns by performing additional experiments and by revising the manuscript. However, some of the key issues were only addressed/discussed in the rebuttal letter with no or minimal changes to the manuscript itself. These key issues as well as some other related issues need to be addressed in the manuscript before publication.

Key issues:

1. Authors believe "changes to ER cholesterol pools regulates autophagy induction" (rebuttal to other relevant issues #3 in my previous comments to the authors), but this point is not clearly communicated in the discussion. Although it remains as speculation, this possibility/model, which is based on various evidences shown in this manuscript, needs to be articulated more clearly in the discussion [e.g., cholesterol levels at autophagosome initiation sites >> cholesterol levels at ER membranes that are associated with autophagosome initiation sites (line 421-422) etc.].
2. Update the figure legend, describing the revised model in Supplementary Figure 7c, by mentioning the possibility that GRAMD1C may also act at ER-PM contacts to limit autophagosome biogenesis (possibly by modulating ER cholesterol levels as the authors stated above); current statement in the figure legend "GRAMD1C localizes to the ER and interacts with mitochondria via its GRAM domain to facilitate cholesterol transport from mitochondria to the ER, thus limiting autophagosome biogenesis" exaggerates the possible role of GRAMD1s at ER-mitochondria contacts in autophagosome biogenesis, and thus, it needs to be toned down. Something like the following statement may be more appropriate: "GRAMD1C localizes to the ER and interacts with mitochondria and the plasma membrane via its GRAM domain to facilitate cholesterol transport from these cellular structures to the ER. Such property of GRAMD1C may contribute to the suppression of autophagosome biogenesis by modulating cholesterol levels at ER membranes that are associated with autophagosome initiation sites"
3. Authors agree that "the results showing a role for GRAMD1C in autophagy and mitochondrial bioenergetic phenotypes in ccRCC cell lines (786-0) are not necessarily directly underlying the survival curves seen for ccRCC patients" (rebuttal to the major issue #3 in the previous comments to the authors). As these data remain rather correlative, this statement and various other statements, including "the mechanisms behind this are most likely more complex than just cell proliferation, changes to mitochondrial respiration or autophagy", need to be clearly mentioned in discussion to avoid misinterpretation (e.g., causality) of the ccRCC survival data and GRAMD1C function at ER-mitochondria contacts.
4. The new data from authors regarding the specificity of mCherry-D4H for detecting the levels of cholesterol in mitochondria are helpful in interpreting the data (rebuttal to the major issue #2 in the previous comments to the authors). They should be included in the manuscript as supplementary data (in the part related to the line 290-291).
5. Given the difficulty in interpreting the data on TSPO knockdown as authors pointed out (rebuttal to the major issue #2 in the previous comments to the authors), TSPO data (supplementary figure 5h-i) should be removed from the current manuscript; the way TSPO depletion affects autophagy maybe entirely different from how GRAMD1C depletion affects autophagy.

Other relevant issues:

1. Discuss in introduction and/or discussion the property of the GRAM domain in the context of its role in cholesterol sensing of GRAMD1s (PMID: 33604931), given the possibility that such cholesterol sensing property, in addition to a potential interaction of the GRAM domain with mitochondrial proteins as shown in the current manuscript, may also contribute to the interaction of GRAMD1C to various membranes (e.g., PM, mitochondria etc.) to regulate autophagy (as suggested in the revised model in Supplementary Figure 7c).
2. "GRAM family" is a quite broad term as there are many more GRAM domain containing proteins other than GRAMD1s, GRAMD2, and GRAMD3 in proteome. Thus, "GRAMD family" or "GRAMD

family proteins" would be more appropriate in the current manuscript.

Several examples are shown below (not exhaustive):

- Line 42 in the abstract: "GRAM family genes" to "GRAMD family proteins".
 - Line 87: "is the GRAM family (consisting of GRAMD1A, GRAMD1B, GRAMD1C, GRAMD2 and GRAMD3, also known as Aster proteins)" to "belongs to the GRAMD family proteins [consisting of GRAMD1s (GRAMD1A, GRAMD1B, and GRAMD1C, also known as Aster proteins), GRAMD2, and GRAMD3]"
 - Line 104-105: "that members of the GRAM family of cholesterol transport proteins" to "that members of the GRAMD family proteins" (not all the GRAMD proteins transport cholesterol; e.g. GRAMD2 and GRAMD3)
 - Line 174-175: "The GRAM family of proteins" to "The GRAMD family proteins"
 - Line 207: "other GRAM family members" to "other GRAMD family proteins"
3. Line 92: "accumulation of cholesterol" should be "accumulation of accessible cholesterol".
4. Briefly describing the origin of A498 cells (line 353) in the main text would be helpful to the readers.

REVIEWERS' COMMENTS

Reviewer #2 (Remarks to the Author):

The authors addressed my major concerns by performing additional experiments and by revising the manuscript. However, some of the key issues were only addressed/discussed in the rebuttal letter with no or minimal changes to the manuscript itself. These key issues as well as some other related issues need to be addressed in the manuscript before publication.

We thank the reviewer for the thorough reading of our manuscript and the insightful comments. We have addressed all comments and concerns as described below and shown in red in the revised manuscript.

Key issues:

1. Authors believe “changes to ER cholesterol pools regulates autophagy induction” (rebuttal to other relevant issues #3 in my previous comments to the authors), but this point is not clearly communicated in the discussion. Although it remains as speculation, this possibility/model, which is based on various evidences shown in this manuscript, needs to be articulated more clearly in the discussion [e.g., cholesterol levels at autophagosome initiation sites >> cholesterol levels at ER membranes that are associated with autophagosome initiation sites (line 421-422) etc.].

We have now corrected the text and the legend to Supplementary Figure 7c (see point below) according to the reviewer’s suggestion. The text now reads (line 418-423) “GRAMD1C localizes to the ER and interacts with mitochondria and the plasma membrane via its GRAM domain to facilitate cholesterol transport from these cellular structures to the ER. Thus, we propose a model where GRAMD1C contributes to the suppression of autophagosome biogenesis by modulating cholesterol levels at ER membranes that are associated with autophagosome initiation sites (Supplementary figure 7c).”

2. Update the figure legend, describing the revised model in Supplementary Figure 7c, by mentioning the possibility that GRAMD1C may also act at ER-PM contacts to limit autophagosome biogenesis (possibly by modulating ER cholesterol levels as the authors stated above); current statement in the figure legend “GRAMD1C localizes to the ER and interacts with mitochondria via its GRAM domain to facilitate cholesterol transport from mitochondria to the ER, thus limiting autophagosome biogenesis” exaggerates the possible role of GRAMD1s at ER-mitochondria contacts in autophagosome biogenesis, and thus, it needs to be toned down. Something like the following statement may be more appropriate: “GRAMD1C localizes to the ER and interacts with mitochondria and the plasma membrane via its GRAM domain to facilitate cholesterol transport from these cellular structures to the ER. Such property of GRAMD1C may contribute to the suppression of autophagosome biogenesis by modulating cholesterol levels at ER membranes that are associated with autophagosome initiation sites”

We have now updated the legend to Supplementary Figure 7c in line with the reviewer’s suggestion.

3. Authors agree that “the results showing a role for GRAMD1C in autophagy and mitochondrial bioenergetic phenotypes in ccRCC cell lines (786-0) are not necessarily directly underlying the survival curves seen for ccRCC patients” (rebuttal to the major issue #3 in the previous comments to the authors). As these data remain rather correlative, this statement and various other statements,

including “the mechanisms behind this are most likely more complex than just cell proliferation, changes to mitochondrial respiration or autophagy”, need to be clearly mentioned in discussion to avoid misinterpretation (e.g., causality) of the ccRCC survival data and GRAMD1C function at ER-mitochondria contacts.

We completely agree with the reviewer and have now included the following sentence to the discussion (line 471-473); “However, the correlation between GRAMD expression levels and patient overall survival is likely more complex than just cell proliferation, changes to mitochondrial respiration or autophagy.”

4. The new data from authors regarding the specificity of mCherry-D4H for detecting the levels of cholesterol in mitochondria are helpful in interpreting the data (rebuttal to the major issue #2 in the previous comments to the authors). They should be included in the manuscript as supplementary data (in the part related to the line 290-291).

We have now included this data as Supplementary figure 4f (referred to in text line 291).

5. Given the difficulty in interpreting the data on TSPO knockdown as authors pointed out (rebuttal to the major issue #2 in the previous comments to the authors), TSPO data (supplementary figure 5h-i) should be removed from the current manuscript; the way TSPO depletion affects autophagy maybe entirely different from how GRAMD1C depletion affects autophagy.

The TSPO knock down was done to investigate if inhibition of another mitochondrial cholesterol transport protein also caused an autophagy phenotype. However, as we are unable to attribute this phenotype to inhibited mitochondrial cholesterol transport (or possibly an accumulation of cholesterol in the OMM), we agree that it is better to remove the data. This has now been done.

Other relevant issues:

1. Discuss in introduction and/or discussion the property of the GRAM domain in the context of its role in cholesterol sensing of GRAMD1s (PMID: 33604931), given the possibility that such cholesterol sensing property, in addition to a potential interaction of the GRAM domain with mitochondrial proteins as shown in the current manuscript, may also contribute to the interaction of GRAMD1C to various membranes (e.g., PM, mitochondria etc.) to regulate autophagy (as suggested in the revised model in Supplementary Figure 7c).

We have now extended our discussion of the GRAM domain in the discussion (line 406-411) and modified the text in the introduction (line 86-89) to read “An example of such cholesterol transport proteins belongs to the GRAMD family proteins (consisting of GRAMD1A, GRAMD1B, GRAMD1C (also known as Aster Proteins), GRAMD2 and GRAMD3), named after the lipid-binding PH-like GRAM domain in their N-terminal region.”

2. “GRAM family” is a quite broad term as there are many more GRAM domain containing proteins other than GRAMD1s, GRAMD2, and GRAMD3 in proteome. Thus, “GRAMD family” or “GRAMD family proteins” would be more appropriate in the current manuscript.

We agree with the reviewer and have now replaced “GRAM family” with “GRAMD family” throughout the text, including the places mentioned below.

Several examples are shown below (not exhaustive):

- Line 42 in the abstract: “GRAM family genes” to “GRAMD family proteins”.

Done.

- Line 87: “is the GRAM family (consisting of GRAMD1A, GRAMD1B, GRAMD1C, GRAMD2 and GRAMD3, also known as Aster proteins)” to “belongs to the GRAMD family proteins [consisting of GRAMD1s (GRAMD1A, GRAMD1B, and GRAMD1C, also known as Aster proteins), GRAMD2, and GRAMD3]”

Thanks, this has now been corrected.

- Line 104-105: “that members of the GRAM family of cholesterol transport proteins” to “that members of the GRAMD family proteins” (not all the GRAMD proteins transport cholesterol; e.g. GRAMD2 and GRAMD3)

This has been changed to “.. that members of the GRAMD1 family of cholesterol transport proteins....” (line 104)

- Line 174-175: “The GRAM family of proteins” to “The GRAMD family proteins”

Corrected (line 174).

- Line 207: “other GRAM family members” to “other GRAMD family proteins”

Corrected (line 207).

3. Line 92: “accumulation of cholesterol” should be “accumulation of accessible cholesterol”.

This has now been changed (line 92).

4. Briefly describing the origin of A498 cells (line 353) in the main text would be helpful to the readers.

We have now included the following text (line 352) “Similar to our observation in U2OS cells, GRAMD1C depletion promoted starvation-induced autophagy in the kidney cell line A498 (Figure 7f-g)”.